# GRAY-BOX GAUSSIAN PROCESSES FOR AUTOMATED REINFORCEMENT LEARNING

**Gresa Shala**[1]**, André Biedenkapp**[1]**, Frank Hutter**[1,2]**, Josif Grabocka**[1]
[1]Department of Computer Science, University of Freiburg
[2]Bosch Center for Artificial Intelligence
{shalag,biedenka,fh,grabocka}@cs.uni-freiburg.de

## ABSTRACT

Despite having achieved spectacular milestones in an array of important real-world applications, most Reinforcement Learning (RL) methods are very brittle concerning their hyperparameters. Notwithstanding the crucial importance of setting the hyperparameters in training state-of-the-art agents, the task of hyperparameter optimization (HPO) in RL is understudied. In this paper, we propose a novel gray-box Bayesian Optimization technique for HPO in RL, that enriches Gaussian Processes with reward curve estimations based on generalized logistic functions. In a very large-scale experimental protocol, comprising 5 popular RL methods (DDPG, A2C, PPO, SAC, TD3), 22 environments (OpenAI Gym: Mujoco, Atari, Classic Control), and 7 HPO baselines, we demonstrate that our method significantly outperforms current HPO practices in RL.

## 1 INTRODUCTION

While Reinforcement Learning (RL) has celebrated amazing successes in many applications (Mnih et al., 2015; Silver et al., 2016; OpenAI, 2018; Andrychowicz et al., 2020; Degrave et al., 2022), it remains very brittle (Henderson et al., 2018; Engstrom et al., 2020). The successes of RL are achieved by leading experts in the field with many years of expertise in the "art" of RL, but the field does not yet provide a technology that broadly yields successes off the shelf. A crucial hindrance for both broader impact and faster progress in research is that an RL algorithm that has been well-tuned for one problem does not necessarily work for another one; especially, optimal hyperparameters are environment-specific and must be carefully tuned in order to yield strong performance.

Despite the crucial importance of strong hyperparameter settings in RL (Henderson et al., 2018; Chen et al., 2018; Zhang et al., 2021; Andrychowicz et al., 2021), the field of hyperparameter optimization (HPO) for RL is understudied. The field is largely dominated by manual tuning, computationally expensive hyperparameter sweeps, or population-based training which trains many agents in parallel that exchange hyperparameters and states (Jaderberg et al., 2017). While these methods are feasible for large industrial research labs, they are costly, substantially increase the $CO_2$ footprint of artificial intelligence research (Dhar, 2020), and make it very hard for smaller industrial and academic labs to partake in RL research. In this paper, we address this gap, developing a computationally efficient yet robust HPO method for RL.

The method we propose exploits the fact that reward curves tend to have similar shapes. As a result, future rewards an agent collects with a given hyperparameter setting can be predicted quite well based on initial rewards, providing a computationally cheap mechanism to compare hyperparameter settings against each other. We combine this insight in a novel gray-boy Bayesian optimization method that includes a parametric reward curve extrapolation layer in a neural network for computing a Gaussian process kernel.

In a large-scale empirical evaluation using 5 popular RL methods (DDPG, A2C, PPO, SAC, TD3), 22 environments (OpenAI Gym: Mujoco, Atari, Classic Control), and 7 HPO baselines, we demonstrate that our resulting method, the *Reward-Curve Gaussian Process (RCGP)*, yields state-of-the-art performance across the board. In summary, our contributions are as follows:

- We introduce a novel method for extrapolating initial reward curves of RL agents with given hyperparameters based on partial learning curves with different hyperparameters.

- We introduce RCGP, a novel Bayesian optimization method that exploits such predictions to allocate more budget to the most promising hyperparameter settings.

- We carry out the most comprehensive experimental analysis of HPO for RL we are aware of to date (including 5 popular RL agents, 22 environments and 8 methods), concluding that RGCP sets a new state of the art for optimizing RL hyperparameters in low compute budgets.

To ensure reproducibility (another issue in modern RL) and broad use of RGCP, all our code is open-sourced at `https://github.com/releaunifreiburg/RCGP`.

## 2 RELATED WORK

RL training pipelines are complex and often brittle (Henderson et al., 2018; Engstrom et al., 2020; Andrychowicz et al., 2021). This makes RL difficult to use for novel applications. To mitigate this, automated reinforcement learning (AutoRL; Parker-Holder et al., 2022) aims to alleviate a human practitioner from the tedious and error prone task of manually setting up the RL pipeline.

While there exists different approaches to automate the choice of algorithm, architecture (Miao et al., 2022) or even environment components (Gleave et al., 2021), in this work we focus on hyperparameter optimization (HPO; Feurer & Hutter, 2019; Bischl et al., 2021) for RL. There exist various approaches in the literature of HPO for RL (see, e.g., Eriksson et al., 2003; Chen et al., 2018; Hertel et al., 2020; Ashraf et al., 2021, for a detailed survey we refer to Parker-Holder et al. (2022)). Due to the non-stationarity of RL training, in recent years, most applications of hyperparameter optimization for RL have focused on dynamically adapting hyperparameters throughout the run. For example, population based training (PBT; Jaderberg et al., 2017) and variants thereof (see, e.g., Franke et al., 2021; Parker-Holder et al., 2020) have found more wide-spread use in the community. This style of HPO uses a population of agents to optimize their hyperparameters while training. Parts of the population are used to explore different hyperparameter settings while the rest are kept to exploit the so far best performing configurations. While this has proven a successful HPO method, a drawback of population based methods is that they come with an increased compute cost due to needing to maintain a population of parallel agents. Thus, most extensions of PBT, such as PB2 (Parker-Holder et al., 2020), aim at reducing the required population size. Still, to guarantee sufficient exploration, larger populations might be required which makes such methods hard to use with small compute budgets.

In the field of automated machine learning (AutoML; Hutter et al., 2019), multi-fidelity optimization has gained popularity to reduce the cost of the optimization procedure. Such methods (see, e.g., Kandasamy et al., 2017; Li et al., 2017; Klein et al., 2017a; Falkner et al., 2018; Li et al., 2020; Awad et al., 2021) leverage lower fidelities, such as dataset subsets, lower number of epochs or low numbers of repetitions, to quickly explore the configuration space. For the special case of number of epochs as a fidelity, there also exists a rich literature on learning curve prediction (Swersky et al., 2014; Domhan et al., 2015; Baker et al., 2017; Chandrashekaran & Lane, 2017; Klein et al., 2017b; Wistuba et al., 2022). Multi-fidelity optimization typically evaluates the most promising configurations on higher fidelities, including the full budget. This style of optimization has proven a cost-efficient way of doing HPO for many applications. Still, multi-fidelity optimization has been explored only little in the context of RL. We are only aware of three such works: Runge et al. (2019) used a multi-fidelity optimizer to tune the hyperparameters of a PPO agent (Schulman et al., 2017) that was tasked with learning to design RNA, allowing the so-tuned agent to substantially improve over the state of the art. Nguyen et al. (2020) also modelled the training curves, providing a signal to guide the search. In the realm of model-based RL, it was shown that dynamic tuning methods such as PBT can produce well-performing policies but often fail to generate robust results whereas static multi-fidelity approaches produced much more stable configurations that might not result in as high final rewards (Zhang et al., 2021). Crucially, however, these previous studies did not evaluate how multi-fidelity and PBT style methods compare in the low budget regime, a setting that is more realistic for most research groups.

## 3  PROPOSED METHOD

**Hyperparameter Optimization (HPO)** focuses on discovering the best hyperparameter configuration $\lambda \in \Lambda$ of a Machine Learning method. Gray-box HPO refers to the concept of cheaper and approximate evaluations of the performance of hyperparameter configurations. For example, we can approximately evaluate the final reward of the configurations of a deep RL method system after every epoch of stochastic gradient descent (gray-box evaluations) (Swersky et al., 2014; Wistuba et al., 2022), instead of waiting for the full convergence (black-box evaluations). The reward after a budget $b$ (i.e. after $b$ epochs) is defined as $R(\lambda, b) : \Lambda \times \mathbb{N} \to \mathbb{R}_+$. To address the noisiness of reward curves of RL algorithms we smooth the curves using a best-so-far transformation. We average rewards based on windows of $h$ training steps, and select the highest reward at any past window, as:

$$R^{(\max)}(\lambda, b) = \max_{0 \le b' < b - h} \frac{1}{h} \sum_{i=1}^{h} R(\lambda, b' + i). \tag{1}$$

From now on, we refer to the smoothed $R^{(\max)}$ as the reward $R$. In addition, let us define the cost (e.g., wall-clock time) of evaluating a configuration for a specific budget as $C(\lambda, b) : \Lambda \times \mathbb{N} \to \mathbb{R}_+$. We define the history of $N$ evaluated configurations and the respective budget as $H^{(K)} = \{(\lambda_1, b_1, R(\lambda_1, b_1)), \ldots, (\lambda_K, b_K, R(\lambda_K, b_K))\}$. A gray-box algorithm $\mathcal{A}$ is a policy that recommends the next configuration to evaluate and its budget as $(\lambda_{K+1}, b_{K+1}) := \mathcal{A}(H^{(K)})$ where $\mathcal{A} : (\Lambda \times \mathbb{N} \times \mathbb{R}_+)^K \to \Lambda \times \mathbb{N}$. Gray-box HPO formally focuses on policies $\mathcal{A}$ that are sequentially executed for as many steps (denoted $K$) as needed to reach a total budget $\Omega$. The best policy discovers the configuration with the largest reward at any budget, as $\arg\max_{\mathcal{A}} \max_{i=1,\ldots,K} R\left((\lambda_{i+1}, b_{i+1}) := \mathcal{A}(H^{(i)})\right)$ s.t. $K = \max_{j \in \mathbb{N}_+} \Omega > \sum_{i=1}^{j} C(\lambda_i, b_i)$.

**Bayesian optimization (BO)** is a very popular HPO policy that sequentially recommends hyperparameters to evaluate. BO operates in sequences of two steps: by (i) fitting a probabilistic regression model to approximate the observed performances $R(\lambda, b)$ of the evaluated configurations and budgets in $H$; and (ii) applying an acquisition to select the next configuration to evaluate.

In the first step, we train Gaussian Processes (Snoek et al., 2012) to approximate the observed performances (i.e. $R(\lambda, b) \approx \mathrm{GP}(\lambda, b; \theta)$) by finding the optimal GP parameters $\theta^*$ via MLE:

$$\theta^*(H) := \arg\max_{\theta} \mathbb{E}_{(\lambda, b, R(\lambda, b)) \sim p_H} \log(R(\lambda, b) \mid \lambda, b; \theta) \tag{2}$$

At the second step, we use an acquisition function, e.g. Expected Improvement (Snoek et al., 2012) $\alpha : \Lambda \times \mathbb{N}_+ \to \mathbb{R}_+$. that scores how "well" a previously unevaluated configuration might perform at a future budget, based on the estimation of the GP fitted above. A naive gray-box BO can be formalized as a special HPO policy, based on a fitted GP with parameters $\theta^*$ from a history of $i$ evaluations $H^{(i)}$, that recommends the $(i + 1)$-th configuration as:

$$(\lambda_{i+1}, b_{i+1}) := \mathcal{A}^{\mathrm{BO}}\left(\theta^*\left(H^{(i)}\right)\right) := \arg\max_{\lambda \in \Lambda, b \in \mathbb{N}_+} \alpha\left(\lambda, b; \theta^*\left(H^{(i)}\right)\right) \tag{3}$$

### 3.1  MULTI-FIDELITY GAUSSIAN PROCESSES WITH REWARD CURVES

A multi-fidelity Gaussian Process can be modeled as a standard GP with an augmented feature vector $z := [\lambda, b] \in \Lambda \times \mathbb{N}$ (Nguyen et al., 2020; Song et al., 2019; Swersky et al., 2013). From the history of evaluations $H^{(K)}$ we define the training features $z_i = [\lambda_i, b_i]$ and their respective targets $y_i = R(\lambda_i, b_i)$ for $i \in \{1, \ldots, K\}$. A kernel function measures the similarity of features as $k(z_i, z_j) : (\Lambda \times \mathbb{N})^2 \to R_+$. The aim of the GP is to estimate the posterior distribution of the unknown target of a new observed test instance $z_* = [\lambda_*, b_*]$. The covariance matrix between training features' pairs is defined as $K(z, z) = [k(z_i, z_j)]_{\forall i,j}$. Similarly, the covariance between test-to-training features is $K(z_*, z) = [k(z_*, z_i)]_{\forall i}$, and the test-to-test one as $K(z_*, z_*) = k(z_*, z_*)$. Ultimately, the posterior prediction of the unknown test target is:

---

**Algorithm 1:** Gray-Box HPO for RL

---

**Input** : Search space $\Lambda$, initial design $H^{(0)}$, budget increment $\Delta_b$, max budget per agent $b^{\max}$
**Output:** Best hyperparameter configuration $\lambda^*$

1 Evaluate initial configurations and budgets $H := H^{(0)}$ ;

2 **while** *still budget* **do**

3      Fit a GP on $H$ using Equations 4-7, for estimating $\mu(\lambda, b), \sigma^2(\lambda, b)$;

4      Run acquisition $a(\mu, \sigma)$ to select $\lambda^{\text{next}} := \arg\max_{\lambda \in \Lambda} a\left(\mu(\lambda, b^{\max}), \sigma^2(\lambda, b^{\max})\right)$;

5      Define the next budget until which to train the selected agent $\lambda^{\text{next}}$:
$$b^{\text{next}} := \min\left(b^{\max}, \begin{cases} \Delta_b & \nexists \lambda^{\text{next}} : (\lambda^{\text{next}}, \cdot, \cdot) \in H \\ \Delta_b + \max_{(\lambda^{\text{next}}, b, \cdot) \in H} b, & \text{otherwise} \end{cases}\right);$$

6      Resume training agent with $\lambda^{\text{next}}$ until $b^{\text{next}}$ and measure reward $R\left(\lambda^{\text{next}}, b^{\text{next}}\right)$;

7      Append to history $H \leftarrow H \cup \{(\lambda^{\text{next}}, b^{\text{next}}, R\left(\lambda^{\text{next}}, b^{\text{next}}\right))\}$;

8 **end**

9 **return** Best configuration $\lambda^*$ with highest reward $\max_{(\lambda^*, b, R(\lambda^*, b)) \in H} R\left(\lambda^*, b\right)$ ;

---

$$\mu\left(z_*\right) = K\left(z_*, z\right)\left(K\left(z, z\right) + \sigma_y^2 I\right)^{-1} y, \tag{4}$$

$$\sigma^2(z_*) = K\left(z_*, z_*\right) - K\left(z_*, z\right)\left(K\left(z, z\right) + \sigma_y^2 I\right)^{-1} K\left(z_*, z\right)^T. \tag{5}$$

It was recently pointed out that a sigmoidal relationship exists between the reward curve of Reinforcement Learning methods and the optimization budget (Nguyen et al., 2020). In this paper, we model the reward curve $R\left(\lambda, b\right)$ of configuration $\lambda$ at budget $b$ as a generalized logistic function (Richard's curve) with five coefficients (Richards, 1959). Furthermore, we do not naively fit one sigmoid function on each reward curve for each hyperparameter configuration. Instead, we propose to condition the sigmoid coefficients on the hyperparameter configurations within a multi-layer perceptron $g : \Lambda \to \mathbb{R}^5$ with weights w (where $g(\lambda, b; w)_i$ represents the $i$-th output neuron) as:

$$\hat{R}\left(\lambda, b; w\right) = g\left(\lambda; w\right)_1 + \frac{g\left(\lambda; w\right)_2 - g\left(\lambda; w\right)_1}{\left(1 + g\left(\lambda; w\right)_3 e^{-g(\lambda; w)_4 b}\right)^{1/g(\lambda; w)_5}}. \tag{6}$$

In this paper, we propose a novel GP that exploits the pattern of the reward curve of the RL algorithm, by introducing the sigmoidal reward curve of (Equation 6). We augment the feature space with the estimation of the reward curve as $[\lambda, b] \to [\lambda, b, \hat{R}\left(\lambda, b; w\right)]$. Therefore, the kernel becomes:

$$k^{\text{our}}\left([\lambda_i, b_i], [\lambda_j, b_j]; w\right) = k\left([\lambda_i, b_i, \hat{R}\left(\lambda_i, b_i; w\right)], [\lambda_j, b_j, \hat{R}\left(\lambda_j, b_j; w\right)]\right). \tag{7}$$

We train the parameters $w$ using the established machinery of kernel learning for GPs (Wilson et al., 2016) and then use this GP for gray-box HPO (Nguyen et al., 2020; Song et al., 2019). We use Expected Improvement (Snoek et al., 2012) at the highest budget $b^{\max}$ as an acquisition function. For brevity, we omit the basics of BO here, however, we refer the interested reader to Snoek et al. (2012). In this context, the reward curve model of Equation 6 offers crucial information in estimating the full ($b^{\max}$) performance of an unknown configuration (i.e., $[\lambda, b^{\max}] \to [\lambda, b^{\max}, \hat{R}\left(\lambda, b^{\max}; w\right)]$), and enables the acquisition function of the BO algorithm to discover performant hyper-parameter configurations. We conducted an analysis on the predictive accuracy of our surrogate in Appendix B.6.

Our novel gray-box HPO method is summarized by the pseudocode of Algorithm 1. We stress that we are concurrently training one agent for each hyperparameter configuration, but we advance

only one training procedure at a time, using an intelligent selection mechanism based on Bayesian optimization. In the first stage, we fit a GP (line 3) using the aforementioned novel kernel function that combines hyperparameter configurations, budgets, and estimated rewards (Equations 4-7). Afterward, we select the next configuration with the highest estimated acquisition at the end of the convergence (line 4). Then we train the RL agent corresponding to the selected configuration for one more budget increment (e.g. continue training for $\Delta_b = 10^b$ more training steps) and measure the observed reward at the end of the next budget (lines 5-6). Note that line 5 defines the next budget for both new configurations ($\not\exists \lambda^{\text{next}} : (\lambda^{\text{next}}, \cdot, \cdot) \in H$) as well as existing ones. We add the evaluation to the history (line 7) and continue the BO procedure until no budget is left (line 3).

## 4 EXPERIMENTAL PROTOCOL

### 4.1 EXPERIMENTAL SETUP

We focus on evaluating the performance of our proposed method, RCGP (Reward-Curve GP), for optimizing the hyperparameters of five popular model-free RL algorithms: PPO (Schulman et al., 2017), A2C Mnih et al. (2016), DDPG (Lillicrap et al., 2016), SAC (Haarnoja et al., 2018), and TD3 (Fujimoto et al., 2018). In total, we consider 22 distinct Gym (Brockman et al., 2016) environments, grouped into the Atari (Bellemare et al., 2013), Classic Control, and Mujoco (Todorov et al., 2012) categories. We denote the full list of environments and their respective action space types in Appendix A, and we list the search spaces for the hyperparameters of each RL algorithm in Table 1.

Table 1: Search spaces for HPO of PPO, A2C, DDPG, SAC, and TD3.

| Algorithm | Hyperparameters | Hyperparameter Values |
|---|---|---|
| PPO | Learning rate ($\log_{10}$) | $\{-6, -5, -4, -3, -2, -1\}$ |
| | $\gamma$ | $\{0.8, 0.9, 0.95, 0.98, 0.99, 1.0\}$ |
| | Clip | $\{0.2, 0.3, 0.4\}$ |
| A2C | Learning rate ($\log_{10}$) | $\{-6, -5, -4, -3, -2, -1\}$ |
| | $\gamma$ | $\{0.8, 0.9, 0.95, 0.98, 0.99, 1.0\}$ |
| DDPG | Learning rate ($\log_{10}$) | $\{-6, -5, -4, -3, -2, -1\}$ |
| | $\gamma$ | $\{0.8, 0.9, 0.95, 0.98, 0.99, 1.0\}$ |
| | $\tau$ | $\{0.0001, 0.001, 0.005\}$ |
| SAC | Learning rate ($\log_{10}$) | $\{-6, -5, -4, -3, -2, -1\}$ |
| | $\gamma$ | $\{0.8, 0.9, 0.95, 0.98, 0.99, 1.0\}$ |
| | $\tau$ | $\{0.0001, 0.001, 0.005\}$ |
| TD3 | Learning rate ($\log_{10}$) | $\{-6, -5, -4, -3, -2, -1\}$ |
| | $\gamma$ | $\{0.8, 0.9, 0.95, 0.98, 0.99, 1.0\}$ |
| | $\tau$ | $\{0.0001, 0.001, 0.005\}$ |

We evaluated static hyperparameter optimization (HPO) methods by querying AutoRL-Bench[1], which is a tabular benchmark for AutoRL that contains reward curves for three different random seeds belonging to runs of RL algorithms with every possible combination of hyperparameter values from the search spaces shown in Table 1. For the dynamic HPO methods (PBT (Jaderberg et al., 2017) and PB2 (Parker-Holder et al., 2020), details in Section 4.2), we ran our own evaluations of the RL pipelines. In every environment, we set the budget for all baselines to the run-time equivalent of 10 full training procedures, based on the expected run-time figures of AutoRL-Bench. Each training procedure consists of $10^6$ steps on the training environment. All methods are evaluated for ten seeds in each environment and RL algorithm. The plots show the mean and standard deviations of the relative ranks of all methods, with the training timesteps in the x-axis. We detail the evaluation protocol in Appendix C and the procedure we use to generate the plots in Appendix D. Furthermore, we included the code for evaluating the performance of all the HPO methods in our GitHub repo https://github.com/releaunifreiburg/RCGP.

---

[1] https://github.com/releaunifreiburg/AutoRL-Bench

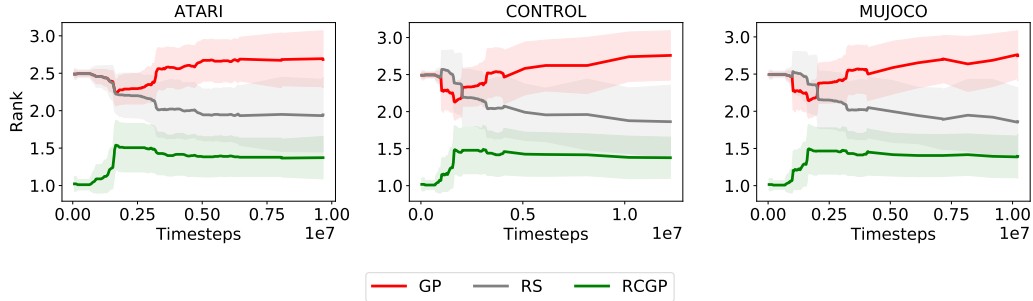

Figure 1: Rank comparison of RS, GP, and RCGP for the **PPO** search space in the Atari, Classic Control, and MuJoCo environments.

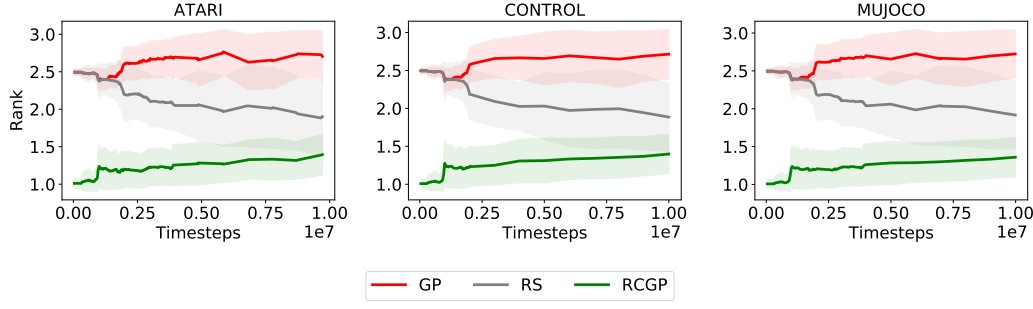

Figure 2: Rank comparison of RS, GP, and RCGP for the **A2C** search space in the Atari, Classic Control, and MuJoCo environments.

## 4.2 BASELINES

We focus on comparing the performance of RCGP to existing HPO approaches within a given time budget. We compare against three types of baselines. The first type includes standard baselines that do not utilize fidelity information during optimization, namely:

- **Random Search (RS)** (Bergstra & Bengio, 2012) is a simple and common HPO baseline. It optimizes hyperparameters by selecting configurations uniformly at random.
- **Bayesian optimization with Gaussian Proccesses (GP)** (Snoek et al., 2012) is another standard HPO baseline, using GPs as the surrogate model in standard blackbox Bayesian optimization. We used a GPytorch (Gardner et al., 2018) implementation with a Matern $5/2$ kernel.

The second type of baselines consists of multi-fidelity baselines which exploit intermediate learning (a.k.a. reward) curve information. Concretely, we compare against the following multi-fidelity HPO techniques:

- **BOHB** (Falkner et al., 2018) is a multi-fidelity HPO baseline that combines Bayesian optimization and Hyperband (Li et al., 2017). It uses tree-based Parzen estimators (TPE) (Bergstra et al., 2011) as a surrogate model for Bayesian optimization. We used the source code provided by the authors.
- **SMAC** (Lindauer et al., 2022) is a recent variant of BOHB that uses Random Forests (RF) as a surrogate model. Here again, we used the implementation provided by the authors.
- **DEHB** (Awad et al., 2021) is a state-of-the-art multi-fidelity HPO baseline that combines Differential Evolution and Hyperband. We used the source code released by the authors.

The third type of baselines includes online HPO techniques (which apply different hyperparameter configurations within a single RL agent training procedure). We compare against two state-of-the-art online HPO methods in RL, that are based on evolutionary search:

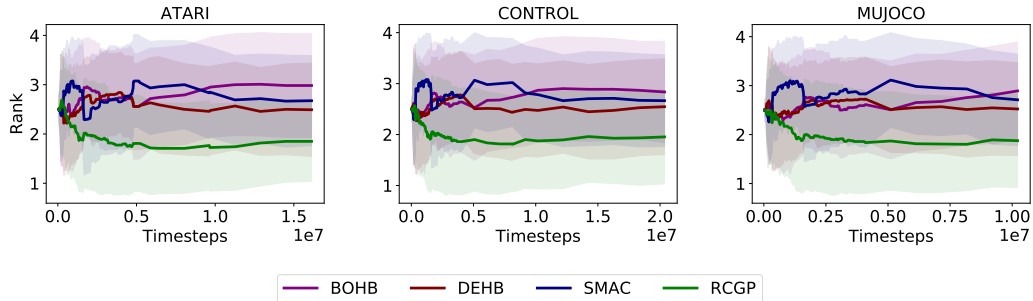

Figure 3: Ranks of BOHB, SMAC, DEHB, and RCGP in Atari, Classic control and MuJoCo enviroments for the **PPO** search space..

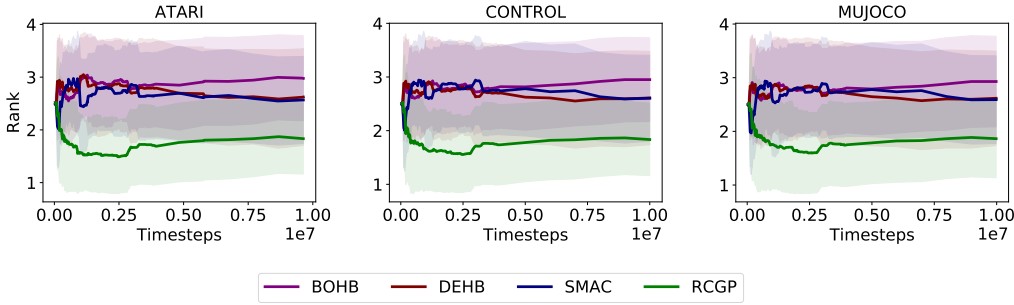

Figure 4: Ranks of BOHB, SMAC, DEHB, and RCGP in Atari, Classic control and MuJoCo enviroments for the **A2C** search space.

- **Population-Based Training (PBT)** (Jaderberg et al., 2017) is an evolutionary HPO method that dynamically optimizes the hyperparameters during the run of the algorithm (i.e., RL agent training). It discards the worst-performing members of the population after a number of steps and replaces them with new hyperparameter configurations that are generated by perturbing the best-performing configuration. We used the PBT implementation in the *Ray Tune* library (Liaw et al., 2018). To facilitate a fair comparison on a small compute budget we follow the protocol of Parker-Holder et al. (2020) and use a population of 4 individuals.

- **Population-based bandits (PB2)** (Parker-Holder et al., 2020) is a PBT-like dynamic HPO method. It replaces the random perturbation with a time-varying GP, as a mechanism to identify well-performing regions of the search space. Again, we used the implementation of PB2 in the *Ray Tune* library (Liaw et al., 2018) with a population of 4 individuals.

## 5 RESEARCH HYPOTHESES AND EXPERIMENTAL RESULTS

**Hypothesis 1**: *Using the reward curve information helps in discovering efficient hyperparameters for model-free RL algorithms in the low budget regime.*

We compare the performance of RCGP to RS and GP, as standard HPO baselines which do not utilize learning curve information. We evaluate RS and GP for 10 full RL algorithm runs. Initially, RS, GP, and RCGP start the search with the same 4 hyperparameter configurations sampled uniformly at random. RCGP queries the learning curve of evaluation returns of these initial configurations for the smallest budget of $10^5$ steps on the training environment. RS, and GP, being black-box optimization methods, query AutoRL-Bench for final evaluation returns after $10^6$ training steps, for both the initial and subsequently suggested configurations. Figure 1 shows the comparison on the PPO search space for the Atari, Classic Control, and Mujoco classes of environments. Similarly, Figure 2 presents the A2C results, while Figure 9 (Appendix B) the results on the DDPG, SAC and TD3 search spaces. In all the cases, RCGP outperforms RS and GP within the wall-clock time budget of our experimental protocol. In Appendix B.2 we also compare to enhanced variants of the black-box HPO methods, which evaluate configurations at smaller budgets. We conclude that gray-box HPO is more efficient than black-box HPO in RL.

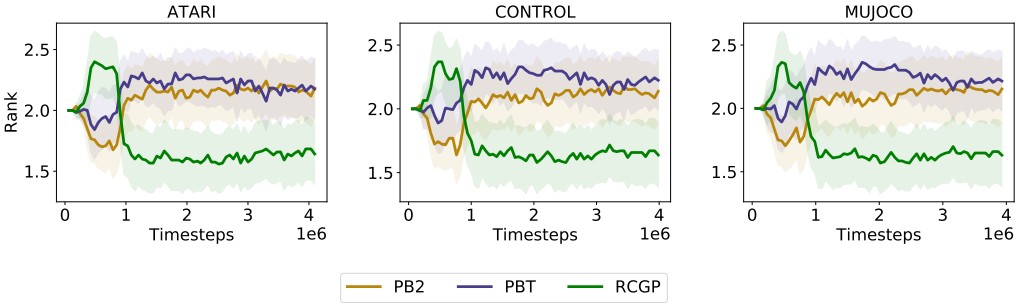

Figure 5: Ranks of PBT, PB2, and RCGP for the **PPO** search space in the Atari, Classic Control, and Mujoco environments.

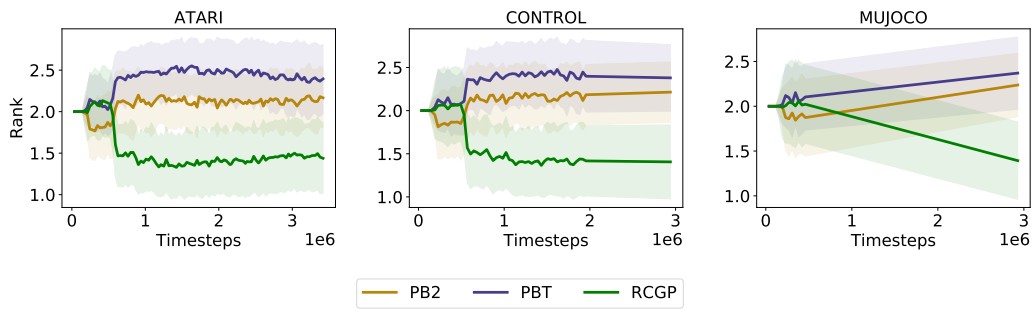

Figure 6: Ranks of PBT, PB2, and RCGP for the **A2C** search space in the Atari, Classic Control, and Mujoco environments.

**Hypothesis 2**: *RCGP outperforms state-of-the-art multi-fidelity HPO methods in optimizing the hyperparameters of model-free RL algorithms.*

We compare the performance of RCGP to BOHB, SMAC and DEHB, as state-of-the-art multi-fidelity HPO baselines. We evaluate each method for the equivalent time of 10 full RL algorithm runs. Initially, all four methods start the search with the same 4 hyperparameter configurations sampled uniformly at random. Figure 3 shows the performance comparison on the PPO search space for the Atari, Classic Control, and Mujoco classes of environments. On the other hand Figure 4 and 10 (Appendix B) include the experiments on the A2C, DDPG, SAC, and TD3 search spaces. In all experiments, our method RCGP on average outperforms BOHB, SMAC, and DEHB in the low budget regime of up to 10 full function evaluations. We therefore conclude that RCGP sets the state-of-the-art in gray-box HPO for RL. We also compared to variants of the multi-fidelity HPO baselines that use max-smoothed reward curves, as detailed in Appendix B.4. In addition, we conducted an analysis of the gain of running the HPO procedure for longer budgets, as detailed in Appendix B.5.

**Hypothesis 3**: *Our method outperforms PBT and PB2, the state-of-the-art HPO in RL.*

In this experiment, we compare our method RCGP to PBT and PB2, to assess the efficiency of our gray-box HPO technique against state-of-the-art dynamic HPO methods. For ensuring a fair comparison, we evaluated PB2 and PBT using the recommended population size of 4 (Parker-Holder et al., 2020), leading to a budget equivalence of 4 full training routines. In addition, all three optimization methods use an initial design consisting of the same 4 hyperparameter configurations sampled uniformly at random. Figure 5 shows the performance comparison on the PPO search space for the Atari, Classic Control, and Mujoco environments. The associated experiments of Figure 6 and 11 (Appendix B), include experiments on A2C, DDPG, SAC, and TD3. The plots show RCGP clearly outperforms PBT and PB2 in the low budget regime of up to 4 full function evaluations. Although PBT and PB2 are able to dynamically configure the hyperparameters of an RL algorithm, they require extensive parallel resources, and thus perform sub-optimally on the low budget regime.

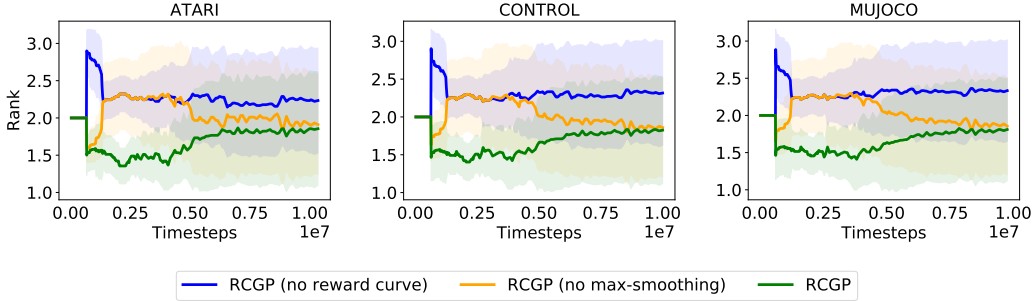

Figure 7: Rank comparison of RCGP (i) without reward curve model, (ii) with raw reward curve information, and (iii) with max-smoothing of the reward curve, for the **PPO** search space in the Atari, Classic Control, and Mujoco environments.

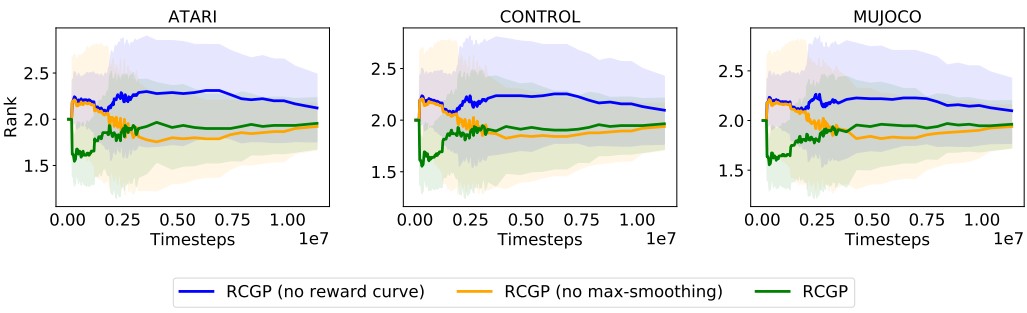

Figure 8: Rank comparison of RCGP (i) without reward curve model, (ii) with raw reward curve information, and (iii) with max-smoothing of the reward curve curve, for the **A2C** search space in the Atari, Classic Control, and Mujoco environments.

## 5.1 ABLATING OUR DESIGN CHOICES

Throughout the paper we based our technical novelty on the hypothesis that reward curves have predictable shapes, and as a result, we can model them accurately with generalized logistic functions (Equation 6). In this section, we ablate the effect of enriching the feature-space of our surrogate with the reward curve estimations, i.e. $[\lambda, b] \rightarrow [\lambda, b, \hat{R}(\lambda, b; w)]$. The ablations of Figure 7-8 demonstrate that using our novel reward curve modeling offers a major boost on the quality of the optimization. In addition, we ablate the effect of the max-smoothing transformation of the reward curves (Equation 1). The empirical results further demonstrate that smoothing the noisy reward curves improves the performance of RCGP in the low-budget regime, especially as the early segments of reward curves are very noisy.

## 6 CONCLUSION

Reinforcement Learning (RL) is one of the premier research sub-areas of Machine Learning, due to the impressive achievements of modern RL methods. Unfortunately, the performance of trained RL agents depends heavily on the choice of the methods' hyperparameters. In this paper we introduced a novel gray-box HPO method that fits Gaussian Processes (GP) to partially-observed reward curves. Our GP variant fuses hyperparameter configurations, budget information and reward curve models based on generalized logistic functions. In a large-scale experimental protocol we demonstrated that our proposed method significantly advances the state-of-the-art for HPO in RL. Especially, we largely outperform evolutionary search HPO methods in RL (PBT and PB2), as well as existing gray-box HPO techniques.

## ACKNOWLEDGEMENTS

We would like to acknowledge the grant awarded by the Eva-Mayr-Stihl Stiftung. In addition, this research was funded by the Deutsche Forschungsgemeinschaft (DFG, German Research Foundation) under grant number 417962828. In addition, Josif Grabocka acknowledges the support of the BrainLinks-BrainTools center of excellence. We acknowledge funding by European Research Council (ERC) Consolidator Grant "Deep Learning 2.0" (grant no. 101045765). Funded by the European Union. Views and opinions expressed are however those of the author(s) only and do not necessarily reflect those of the European Union or the ERC. Neither the European Union nor the ERC can be held responsible for them. The authors acknowledge funding by The Carl Zeiss Foundation through the research network "Responsive and Scalable Learning for Robots Assisting Humans" (ReScaLe) of the University of Freiburg.

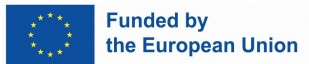

**Reproducibility Statement.** To ensure reproducibility, we give detailed descriptions of the experimental setup including the search spaces, environments, and seeds in Section 4 and Appendix A. We also make available the code we used to generate the results that we present in this paper. The benchmark we use is publicly available, and we reference it correspondingly.

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

## A    RL ENVIRONMENTS

Table 2: List of environments for the experiments.

| Environment Class | Environment Name | Action Space |
| --- | --- | --- |
| Atari | Pong-v0 | Discrete |
| | Alien-v0 | |
| | BankHeist-v0 | |
| | BeamRider-v0 | |
| | Breakout-v0 | |
| | Enduro-v0 | |
| | Phoenix-v0 | |
| | Seaquest-v0 | |
| | SpaceInvaders-v0 | |
| | Riverraid-v0 | |
| | Tennis-v0 | |
| | Skiing-v0 | |
| | Boxing-v0 | |
| | Bowling-v0 | |
| | Asteroids-v0 | |
| Classic Control | CartPole-v1 | Discrete |
| | MountainCar-v0 | |
| | Acrobot-v1 | |
| | Pendulum-v0 | Continuous |
| MuJoCo | Ant-v2 | Continuous |
| | Hopper-v2 | |
| | Humanoid-v2 | |

## B    ADDITIONAL EXPERIMENTAL RESULTS

In this Section we present additional experimental results, including other search spaces (DDPG, SAC, and TD3) in  B.1, comparisons to the black-box HPO baselines modified to run suggested configurations for a smaller budget than the budget of a full RL algorithm training procedure in B.2, comparison to PBT running on a continuous version of the PPO search space in  B.3, as well as average episodic reward plots comparing PBT, PB2, SMAC, BOHB, DEHB, and RCGP in  B.4. We additionally show average normalized regret plots comparing SMAC, BOHB, DEHB, and RCGP in B.5. In  B.6, we evaluate the predictive performance of RCGP with the max-smoothing transformation of the learning curves, and the version trained using non-transformed learning curves.

### B.1    RESULTS ON OTHER SEARCH SPACES

In Figures 9- 11 we show the average rank for the baselines and RCGP across 10 seeds and all MuJoCo environments for the DDPG, SAC, and TD3 search spaces.

### B.2    COMPARISON TO BLACK-BOX BASELINES

In Figures 12- 20 we show the average rank for the black-box HPO baselines with a modified search protocol and RCGP. To allow the black-box baselines to search for more configurations within the maximum search budget, during the search the suggested configurations are evaluated for $\frac{1}{8}$, $\frac{1}{4}$ and $\frac{1}{2}$ of a full RL algorithm training procedure. During plotting, we still adhere to the same plotting protocol as indicated in Algorithm 3.

### B.3    SENSIBILITY OF SEARCH SPACE DISCRETIZATION

Figure 21 shows a comparison of RCGP, PBT optimizing on the discrete search space of PPO (as specified in Table 1), as well as PBT optimizing on a continuous version of this search space, with

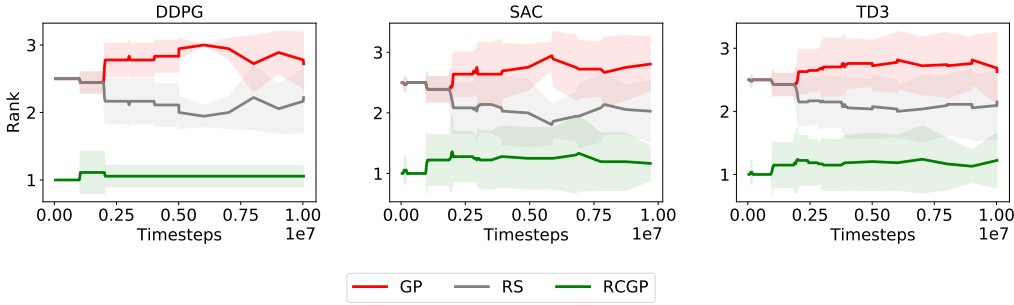

Figure 9: Rank comparison of RS, GP, and RCGP in the MuJoCo enviroments for the **DDPG, SAC, and TD3** search spaces.

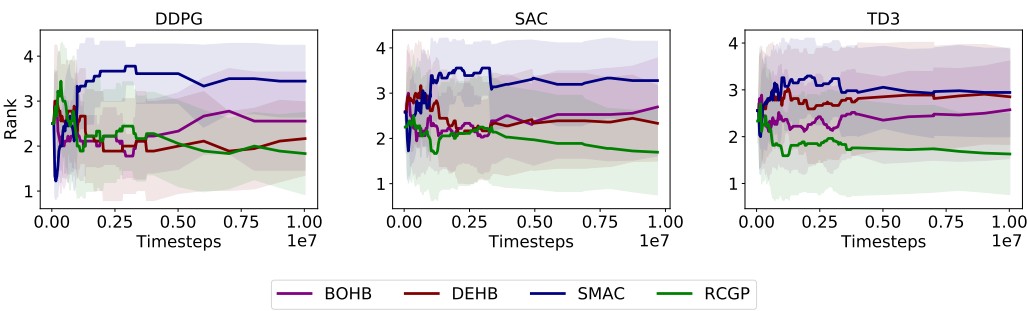

Figure 10: Ranks of BOHB, SMAC, DEHB, and RCGP in the MuJoCo enviroments for the **DDPG, SAC, and TD3** search space.

the same bounds as the discrete version. We label the latter as *PBT-cont*. The performance of PBT is visibly similar compared to PBT-cont on the MUJOCO environments, thus showing the sensibility of the discretization.

### B.4    AVERAGE EPISODIC REWARD PLOTS

In Figures 22- 26, we show the average episodic reward across the search budget for PBT, PB2, and RCGP in each environment for each search space. Figures 27- 31, show the average episodic reward across the search budget for SMAC, BOHB, DEHB, and RCGP in each environment for each search space. All these methods use non-transformed episodic reward curves. We show the comparison in terms of average episodic reward of the grey-box baselines and RCGP using max-smoothed episodic reward curves in Figures 32- 36.

### B.5    NORMALIZED REGRET PLOTS

A common metric in the HPO community for evaluating the distance of the discovered configuration (a.k.a. incumbent) to the optimal configuration (a.k.a. oracle) is the regret defined as (oracle - incumbent). If we let the HPO run longer we are going to discover the oracle in the best-case scenario. In our tabular benchmark, the oracle is known (we can argmax the final returns of all possible hyperparameter configurations), therefore we can directly assess the potential gain of running longer HPO by computing the regret. Furthermore, a percentual difference variant is (oracle - incumbent)/oracle, which shows the potential gain in percentage.

We show the performance of SMAC, BOHB, DEHB, and RCGP in terms of average percentual regret for each environment in all search spaces in Figures 37- 41. All four methods use max-smoothed episodic reward curves. As can be observed the percentual regret is between $10\%$ and $1\%$.

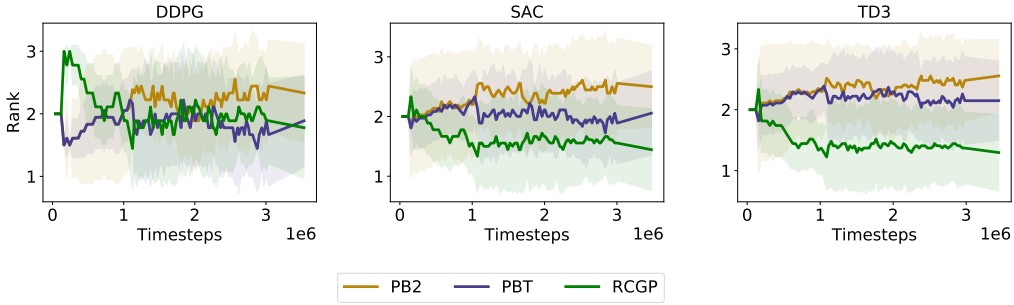

Figure 11: Ranks of PBT, PB2, and RCGP in the Mujoco enviroments for the **TD3** search space.

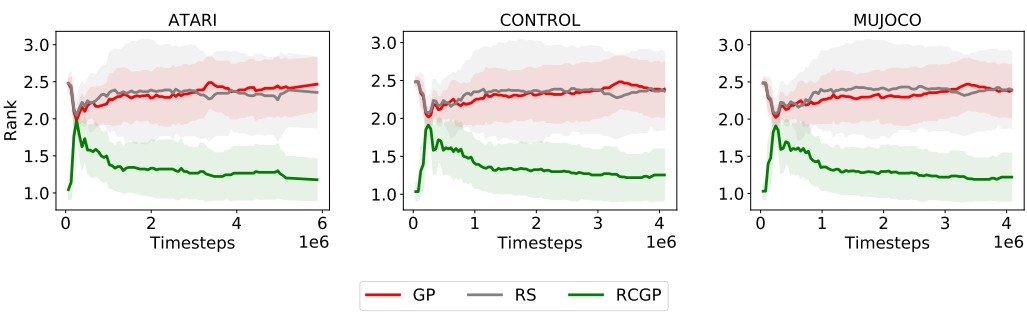

Figure 12: Average ranks of RS, GP, and RCGP in each environment for the **PPO** search space. RS and GP evaluate each suggested configuration during the search for $\frac{1}{8}$ of the full budget ($12 \cdot 10^4$ training timesteps).

Therefore, letting our HPO run till infinity will improve the results between $1\%$ to $10\%$ in the best-case scenario..

### B.6 PREDICTIVE PERFORMANCE OF RCGP

In Figure 42 we show the predictive performance, and the rank correlation of RCGP. The plots show as y-axis (i) the predictive error (square error) and (ii) the Pearson correlation. The x-axis indicates the length of the reward curves which we train RCGP on, in order to estimate the final return. The ground-truth final return is the final return of the non-max-smoothed reward curve.

In terms of predictive accuracy, the plots show that the error decreases as a longer segment of the reward curve is observed. The squared error of RCGP using max-smoothed curves for training is higher than the version using the original reward curve to train. However, in terms of the correlation of ranks of the predicted vs. the ground truth values of the final return, RCGP with max-smoothing has a significantly larger correlation between estimated and final return values. As the correlation is more essential for HPO than the forecasting loss, these plots provide a strong analysis explaining the superiority of our method in terms of HPO performance.

## C EVALUATION PROTOCOL

We detail the evaluation protocol we use for all baselines and RCGP in Algorithm 2.

## D PLOTTING PROTOCOL

The detailed procedure that we use to generate the plots showing the rank of the methods is shown in Algorithm 3.

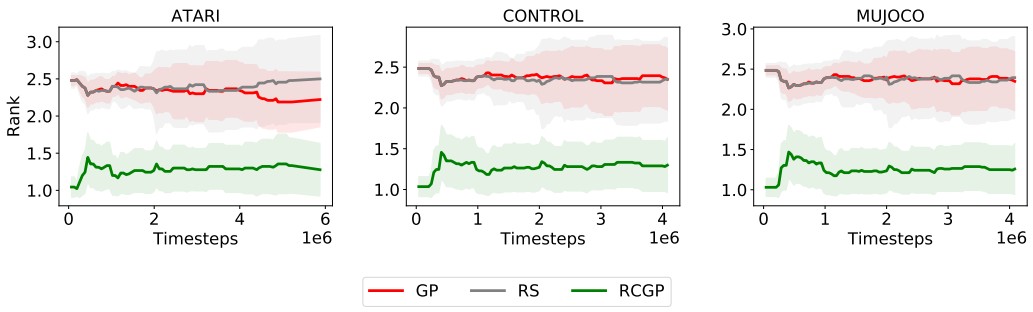

Figure 13: Average ranks of RS, GP, and RCGP in each environment for the **PPO** search space. RS and GP evaluate each suggested configuration during the search for $\frac{1}{4}$ of the full budget ($25 \cdot 10^4$ training timesteps).

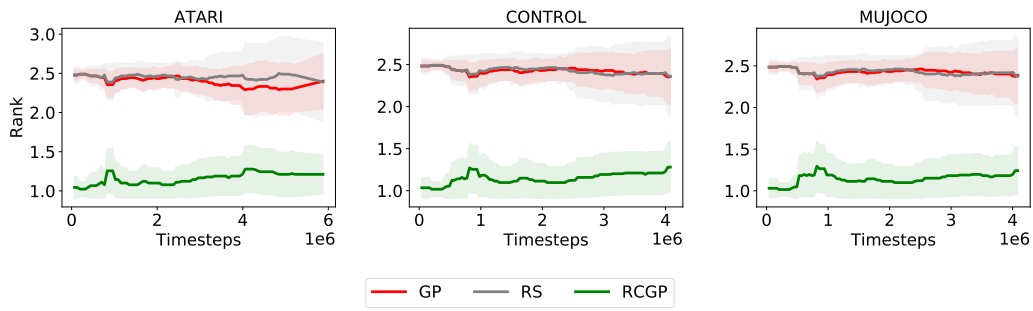

Figure 14: Average ranks of RS, GP, and RCGP in each environment for the **PPO** search space. RS and GP evaluate each suggested configuration during the search for $\frac{1}{2}$ of the full budget ($50 \cdot 10^4$ training timesteps).

---

**Algorithm 2:** Evaluation protocol

1  **for** seed *in SEEDS* **do**
2     **for** *different budgets B* **do**
3        1. Search phase:
4          Run HPO for a budget B and return the best hyperparameter configuration.
5        2. Evaluation phase:
6          Take the best configuration returned in step 1.
7          Train it for the full budget.
8          Evaluate it for 10 episodes and output the mean final return of these episodes.
9     **end**
10 **end**

---

# E   AUTORL-BENCH REWARD CURVES

To further motivate our design choice for modeling the reward curves using a generalised logistic function, we have plotted the reward curves in the AutoRL-Bench tabular benchmark. Figures 43- 47 show the reward curves for the PPO, A2C, DDPG, SAC, and TD3 search spaces, respectively.

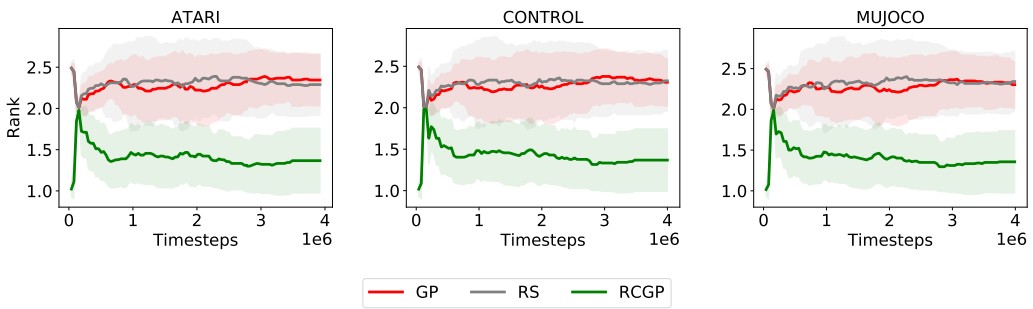

Figure 15: Average ranks of RS, GP, and RCGP in each environment for the **A2C** search space. RS and GP evaluate each suggested configurations during the search for $\frac{1}{8}$ of the full budget ($12 \cdot 10^4$ training timesteps).

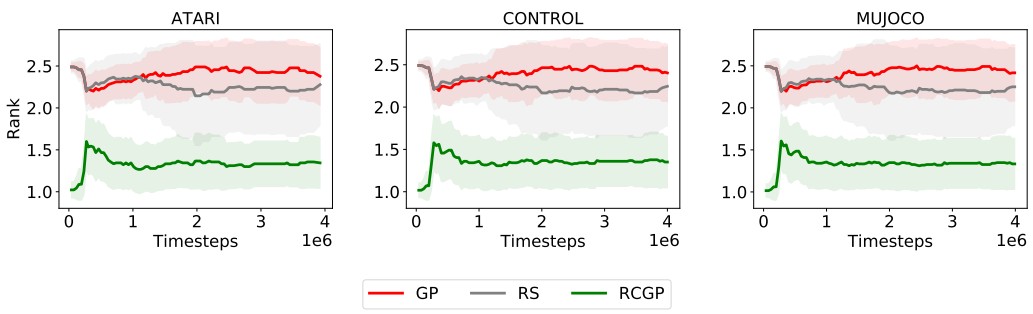

Figure 16: Average ranks of RS, GP, and RCGP in each environment for the **A2C** search space. RS and GP evaluate each suggested configurations during the search for $\frac{1}{4}$ of the full budget ($25 \cdot 10^4$ training timesteps).

---

**Algorithm 3:** Plotting protocol

---

1 **for** seed *in SEEDS* **do**
2     **for** *different budgets B in the* x-*axis* **do**
3         Compute the y-axis by following steps (i) and (ii):
4         (i)   **for** *each environment in a benchmark* **do**
5             **for** *each method* **do**
6                 Get final return (step 2 in Algorithm 2).
7             **end**
8             Compute the rank of each method for that environment based on the returns.
9         **end**
10         (ii)   Compute the mean of the ranks across the environments in step (i) for each method.
11     **end**
12 **end**

---

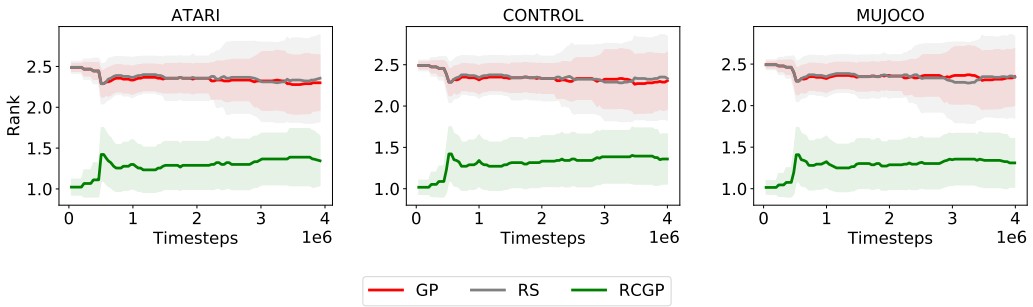

Figure 17: Average ranks of RS, GP, and RCGP in each environment for the **A2C** search space. RS and GP evaluate each suggested configurations during the search for $\frac{1}{2}$ of the full budget ($50 \cdot 10^4$ training timesteps).

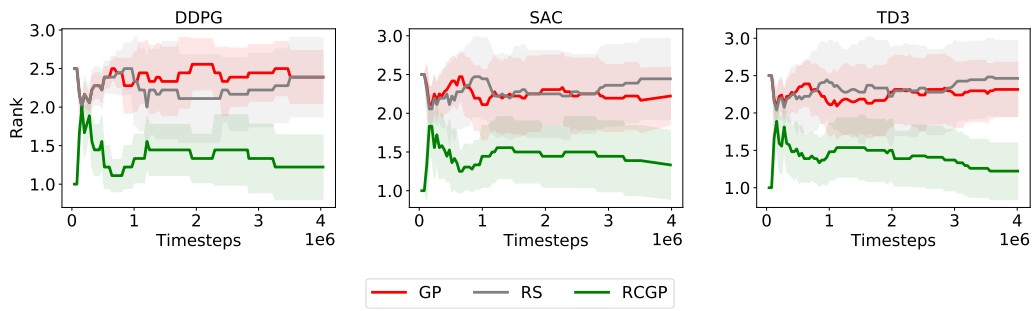

Figure 18: Average ranks of RS, GP, and RCGP in the MuJoCo enviroments for the **DDPG**, **SAC**, and **TD3** search spaces. RS and GP evaluate each suggested configurations during the search for $\frac{1}{8}$ of the full budget ($12 \cdot 10^4$ training timesteps).

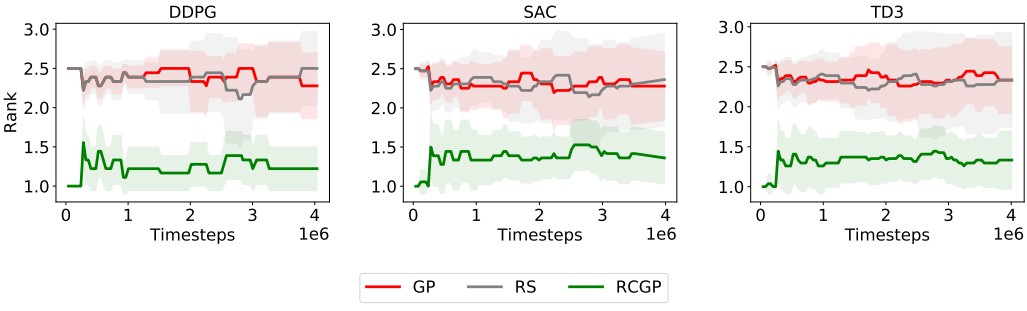

Figure 19: Average ranks of RS, GP, and RCGP in the MuJoCo enviroments for the **DDPG**, **SAC**, and **TD3** search spaces. RS and GP evaluate each suggested configurations during the search for $\frac{1}{4}$ of the full budget ($25 \cdot 10^4$ training timesteps).

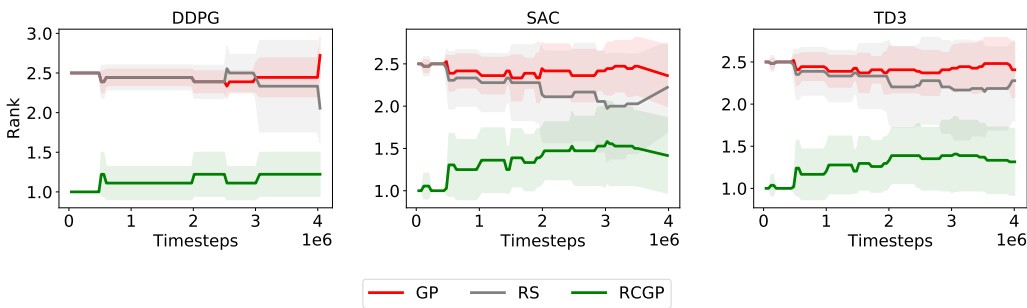

Figure 20: Average ranks of RS, GP, and RCGP in the MuJoCo enviroments for the **DDPG**, **SAC**, and **TD3** search spaces. RS and GP evaluate each suggested configurations during the search for $\frac{1}{2}$ of the full budget ($50 \cdot 10^4$ training timesteps).

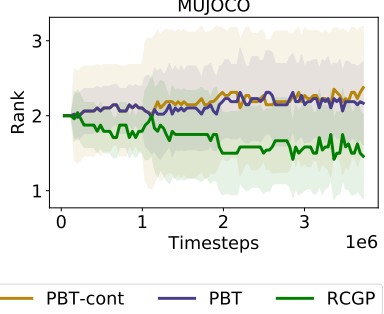

Figure 21: Ranks of RCGP, PBT (optimizing in the discrete search space of AutoRL-Bench), and PBT-cont (optimizing in a continuous search space with the same boundaries as the AutoRL-Bench search space) in the MuJoCo enviroments for the **PPO** search space.

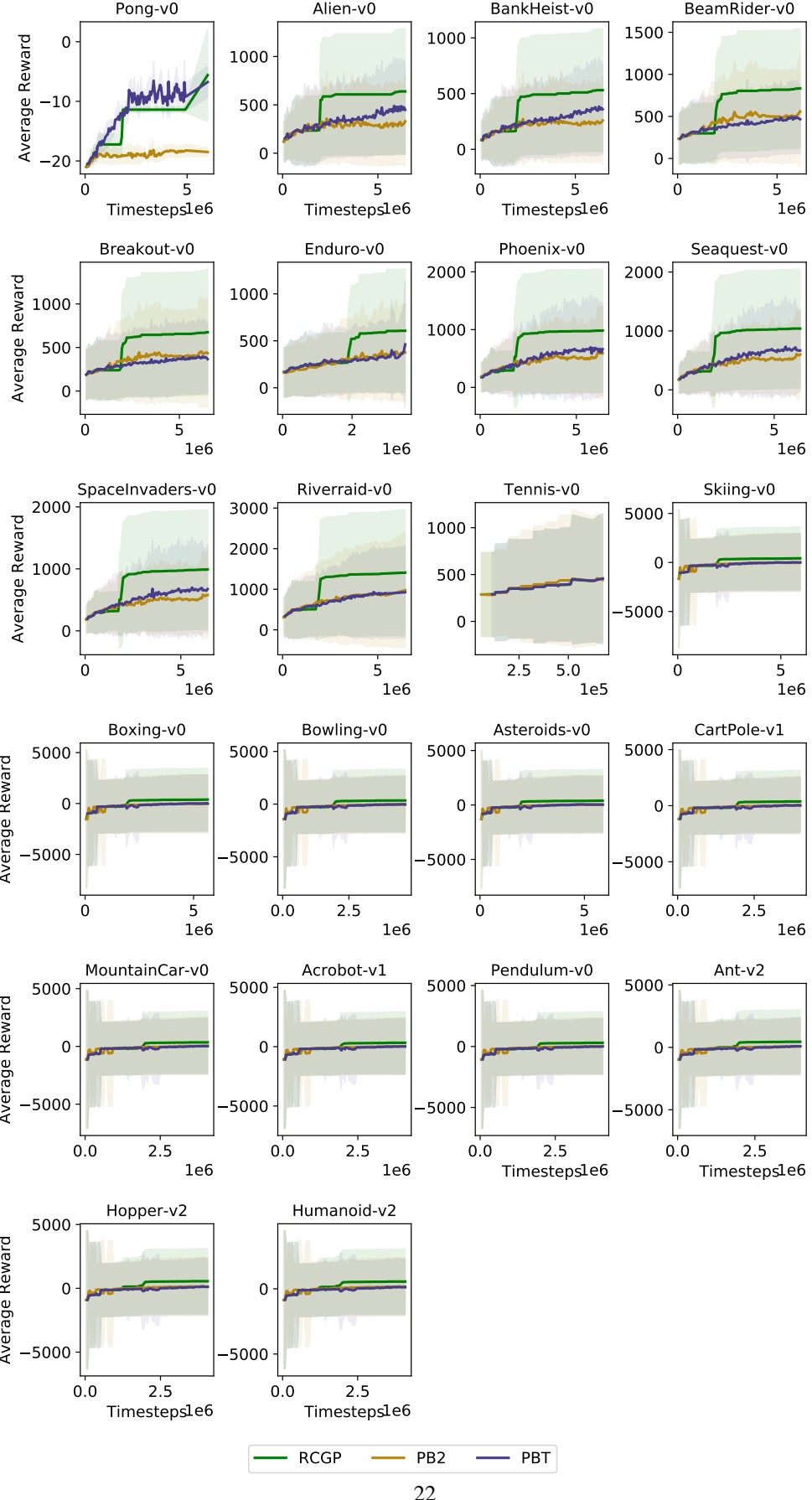

Figure 22: Average reward of PBT, PB2, and RCGP in each enviroment for the **PPO** search space.

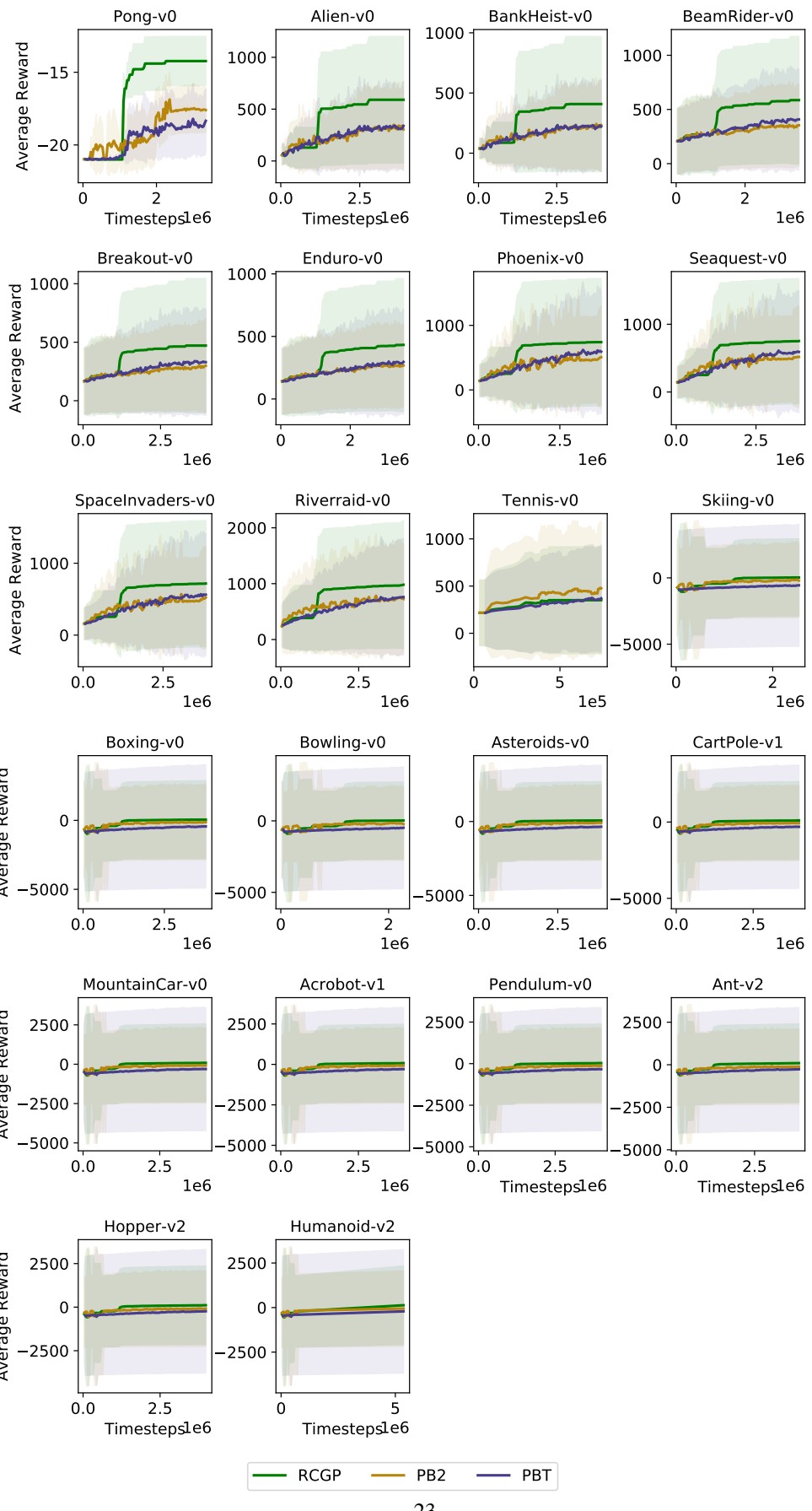

Figure 23: Average reward of PBT, PB2, and RCGP in each enviroment for the **A2C** search space.

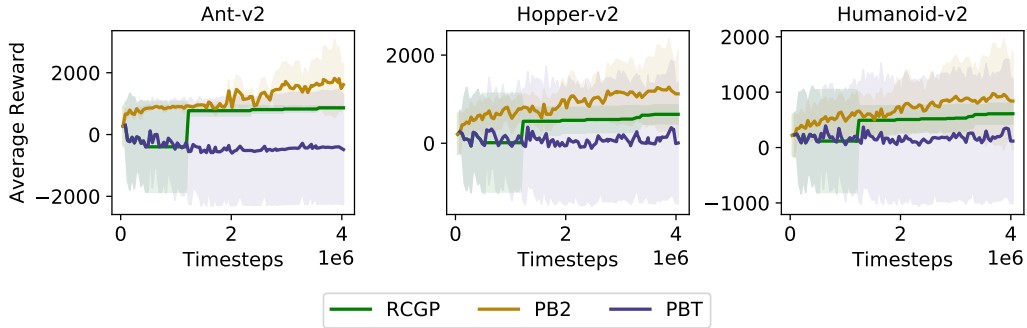

Figure 24: Average reward of PBT, PB2, and RCGP in each of the MuJoCo enviroments for the **DDPG** search space.

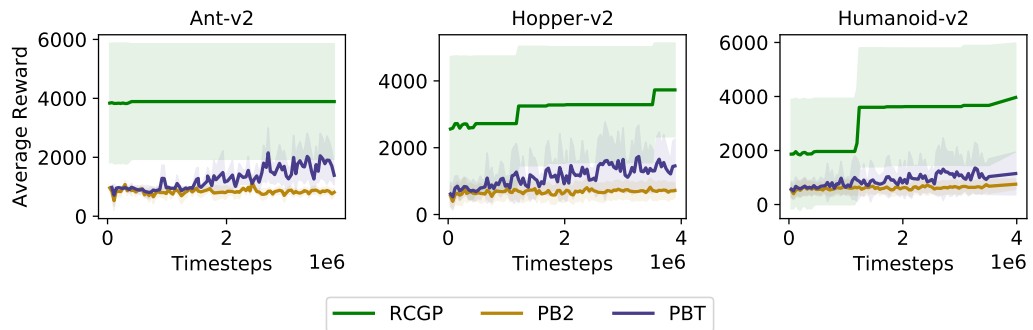

Figure 25: Average reward of PBT, PB2, and RCGP in each of the MuJoCo enviroments for the **SAC** search space.

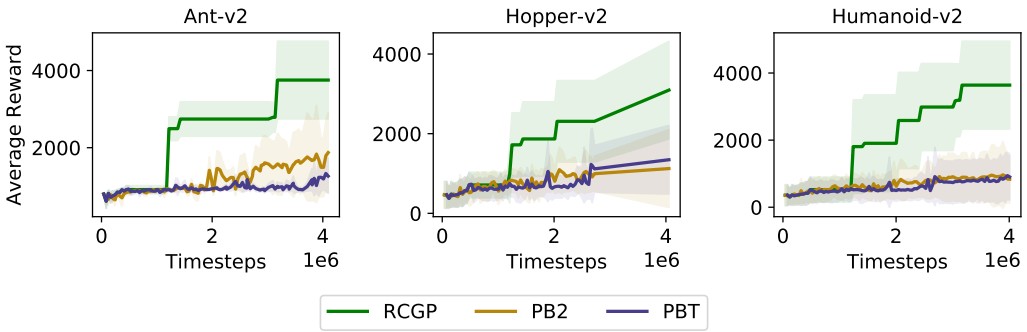

Figure 26: Average reward of PBT, PB2, and RCGP in each of the the MuJoCo enviroments for the **TD3** search space.

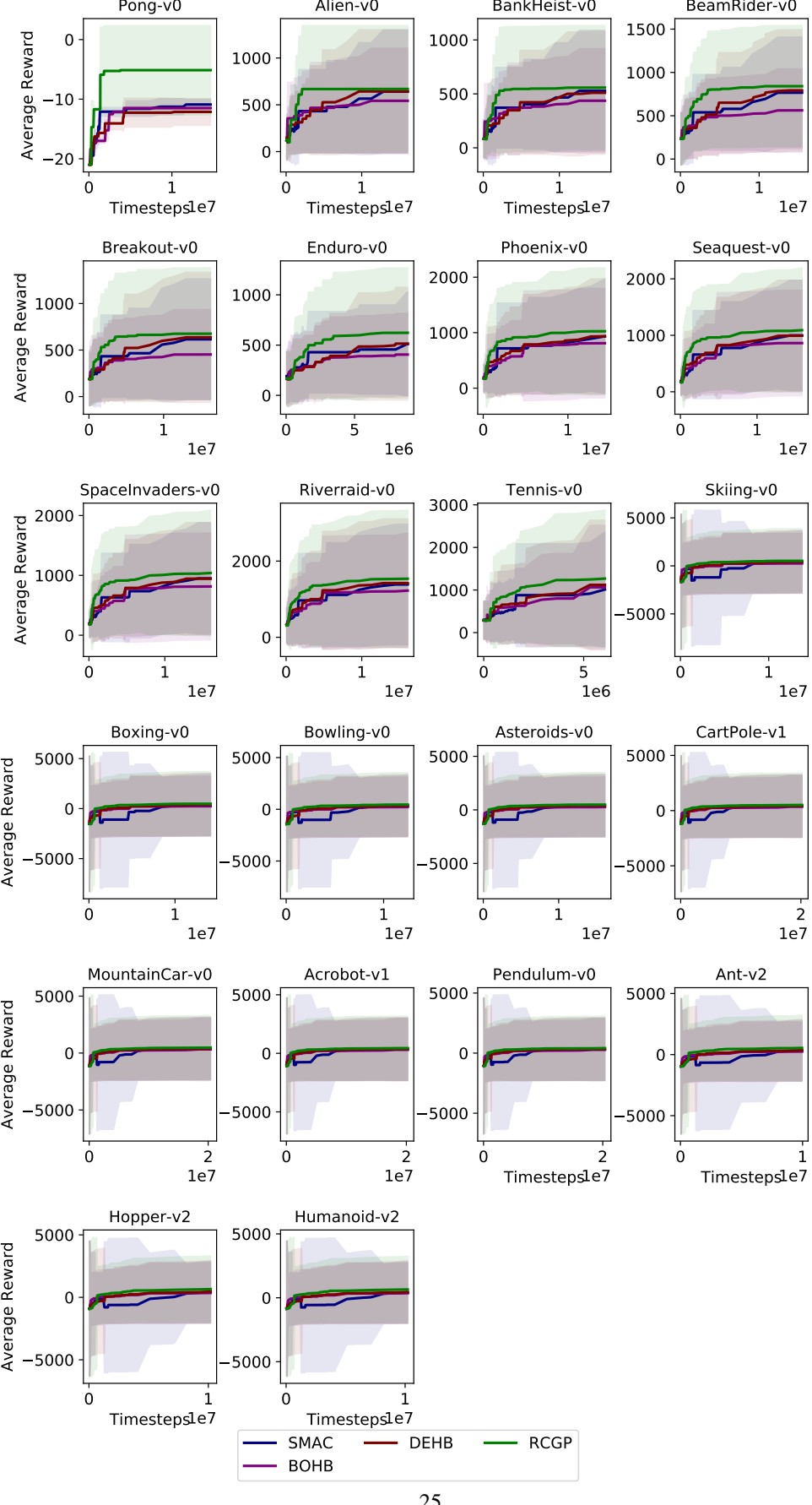

Figure 27: Average reward of SMAC, BOHB, DEHB, and RCGP in each environment for the **PPO** search space using non-transformed episodic reward curves.

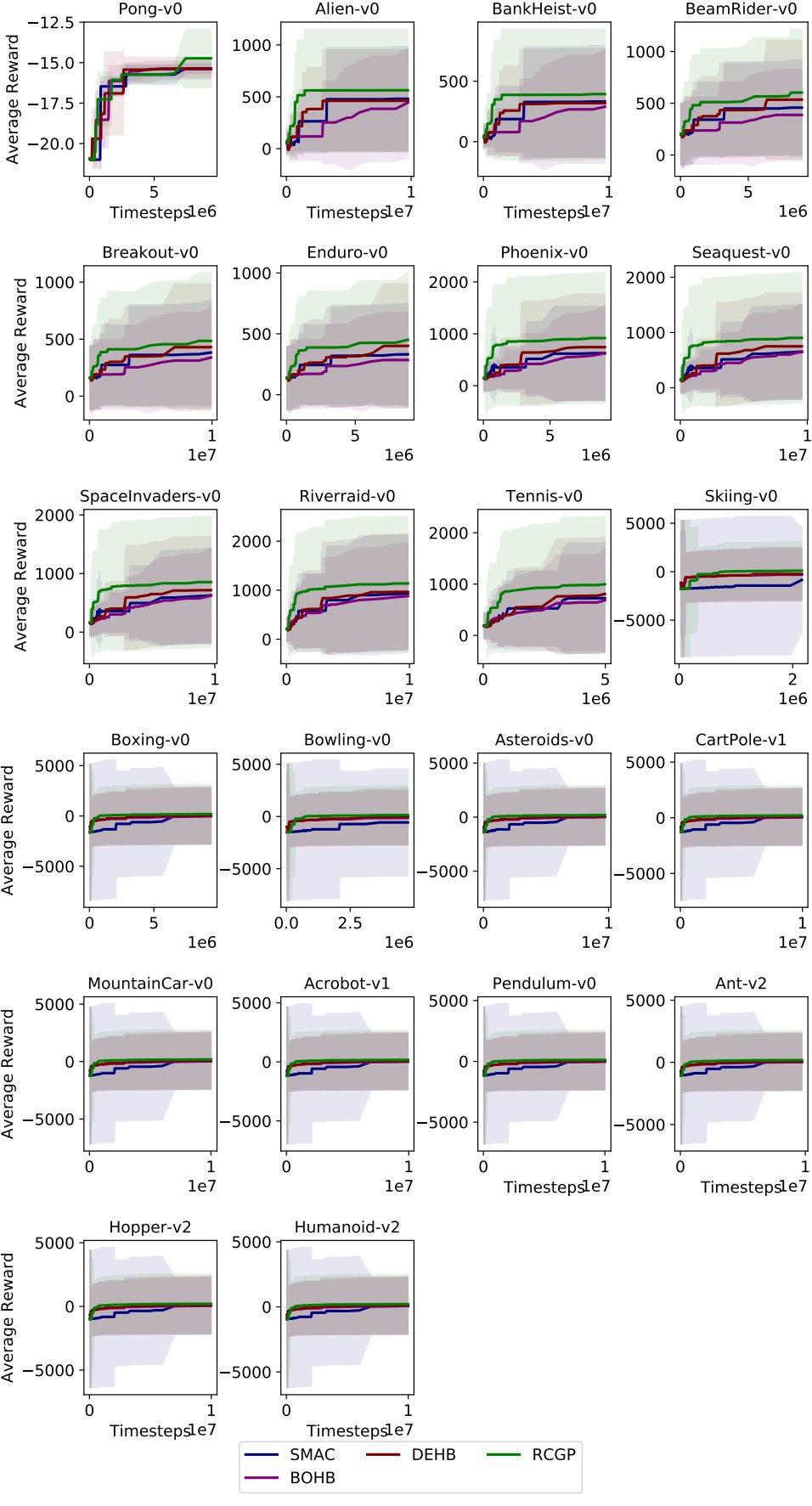

Figure 28: Average reward of SMAC, BOHB, DEHB, and RCGP in each environment for the **A2C** search space using non-transformed episodic reward curves.

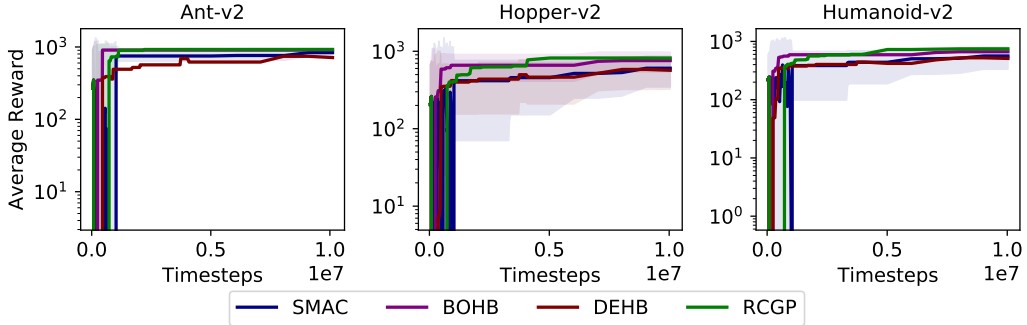

Figure 29: Average reward of SMAC, BOHB, DEHB, and RCGP in each of the MuJoCo enviroments for the **DDPG** search space using non-transformed episodic reward curves.

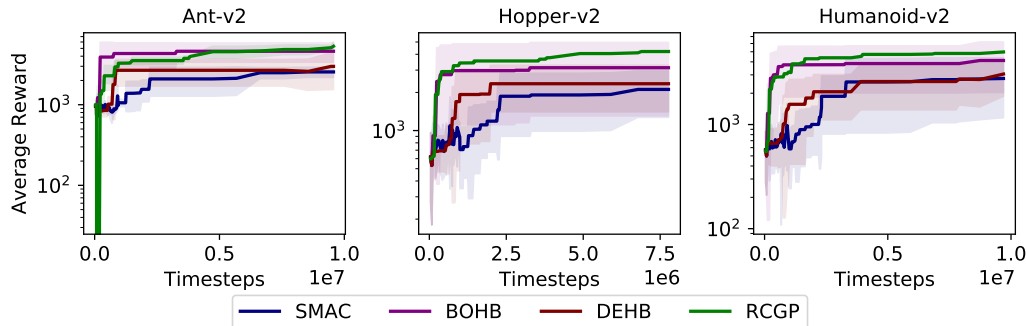

Figure 30: Average reward of SMAC, BOHB, DEHB, and RCGP in each of the MuJoCo enviroments for the **SAC** search space using non-transformed episodic reward curves.

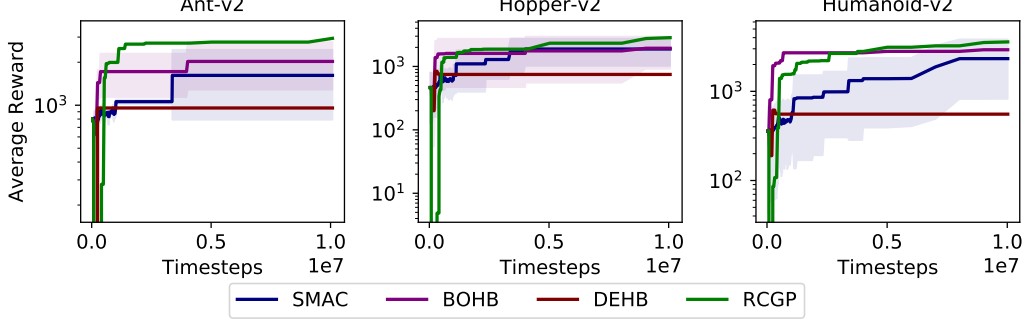

Figure 31: Average reward of SMAC, BOHB, DEHB, and RCGP in each of the the MuJoCo enviroments for the **TD3** search space using non-transformed episodic reward curves.

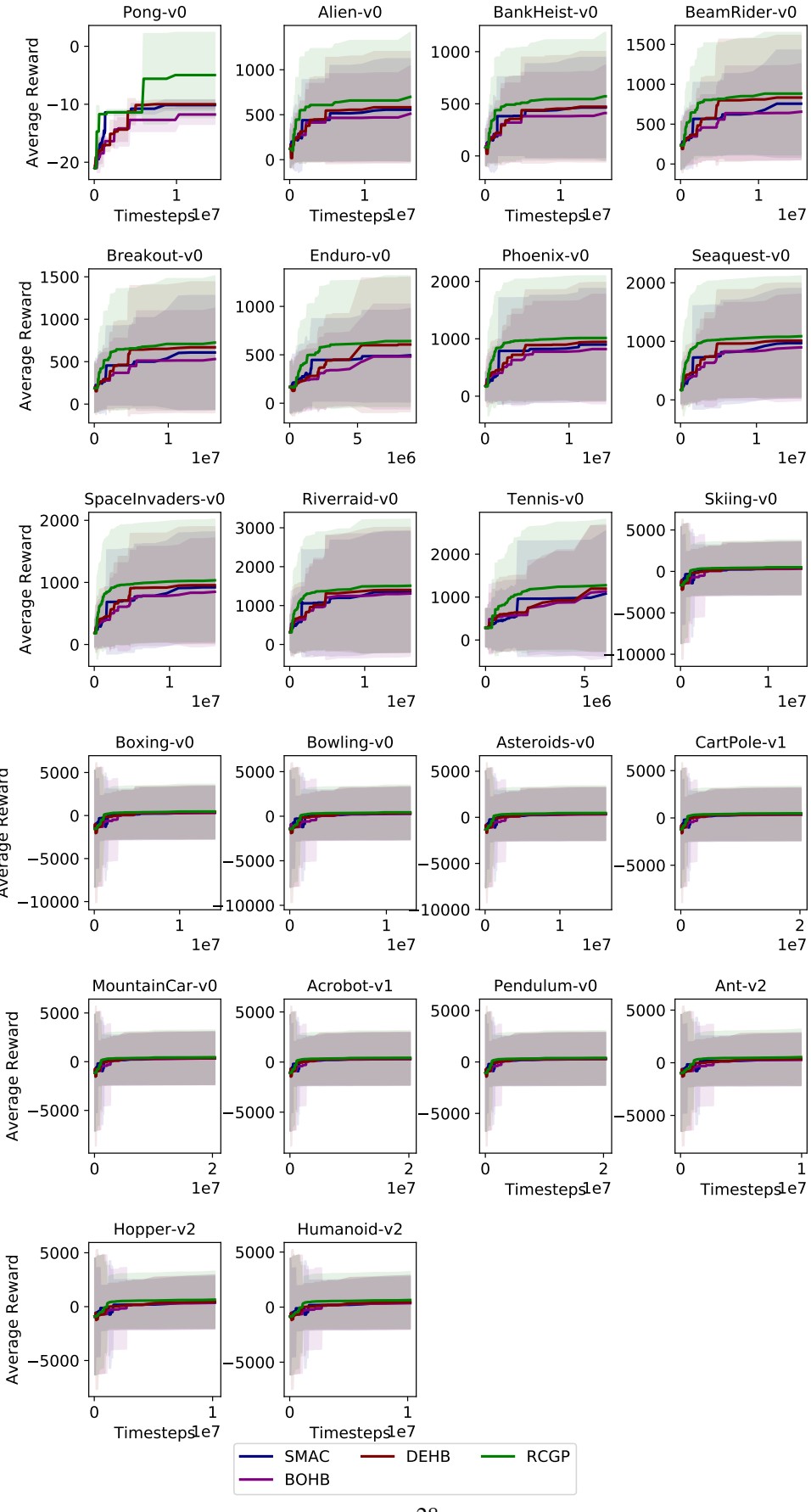

Figure 32: Average reward of SMAC, BOHB, DEHB, and RCGP in each environment for the **PPO** search space using max-smoothed episodic reward curves.

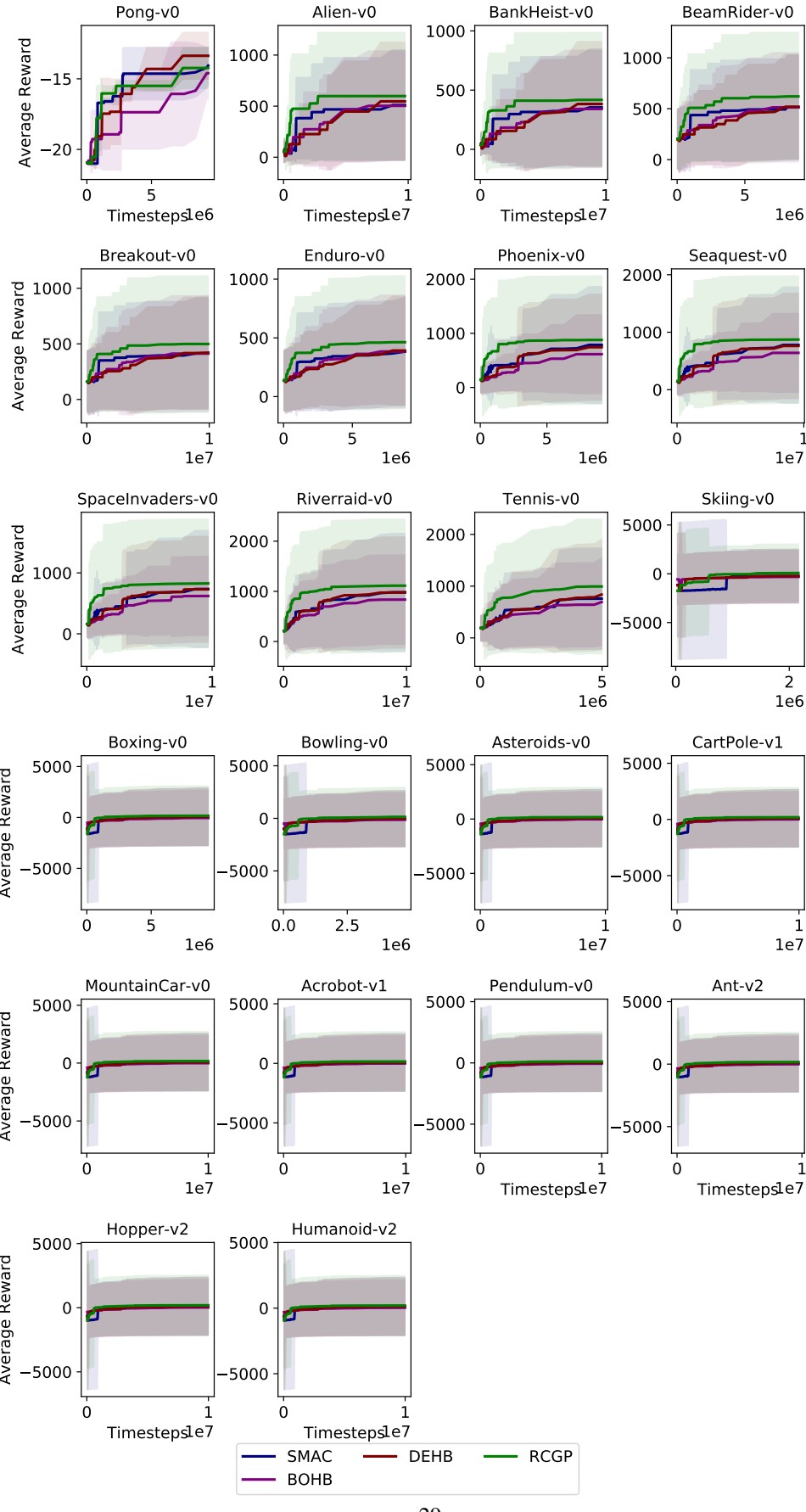

Figure 33: Average reward of SMAC, BOHB, DEHB, and RCGP in each environment for the **A2C** search space using max-smoothed episodic reward curves.

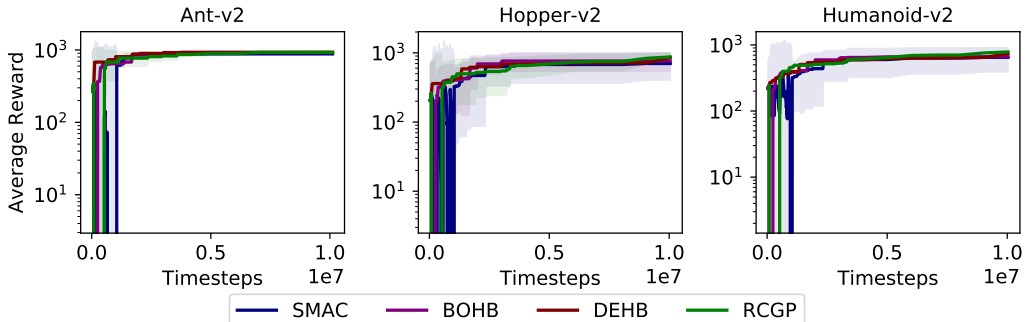

Figure 34: Average reward of SMAC, BOHB, DEHB, and RCGP in each of the MuJoCo enviroments for the **DDPG** search space using max-smoothed episodic reward curves.

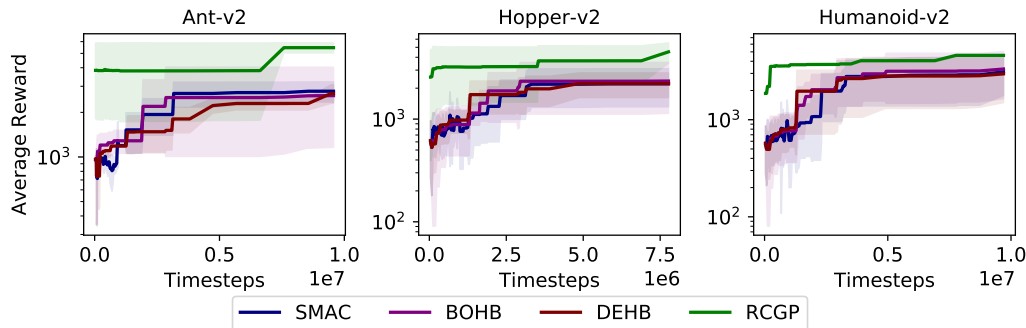

Figure 35: Average reward of SMAC, BOHB, DEHB, and RCGP in each of the MuJoCo enviroments for the **SAC** search space using max-smoothed episodic reward curves.

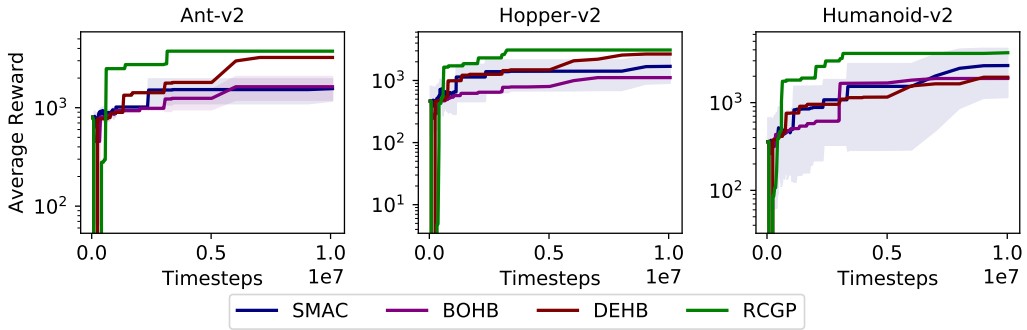

Figure 36: Average reward of SMAC, BOHB, DEHB, and RCGP in each of the the MuJoCo enviroments for the **TD3** search space using max-smoothed episodic reward curves.

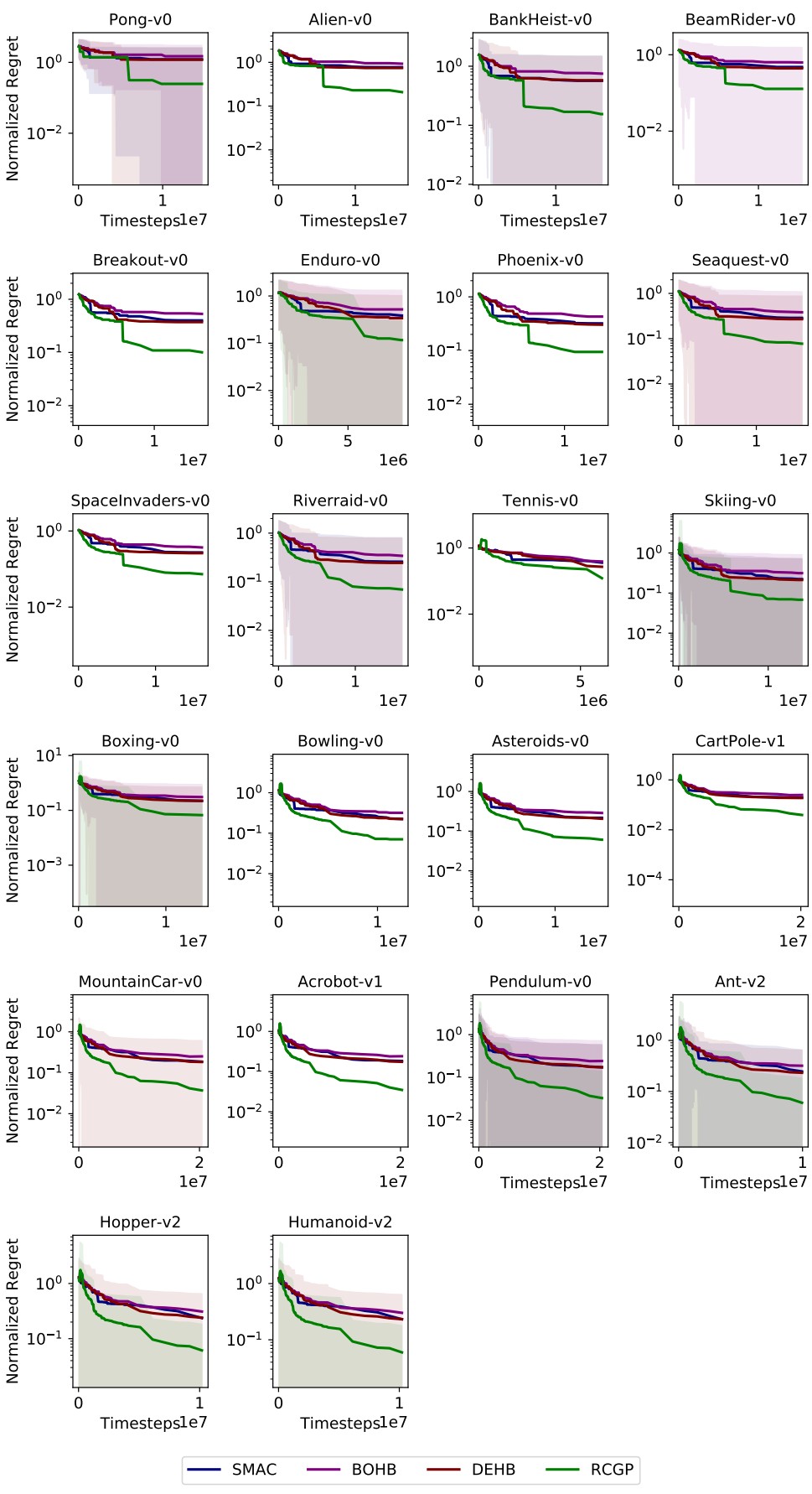

Figure 37: Performance in terms of normalized regret for SMAC, BOHB, DEHB, and RCGP in each environment for the **PPO** search space using max-smoothed episodic reward curves.

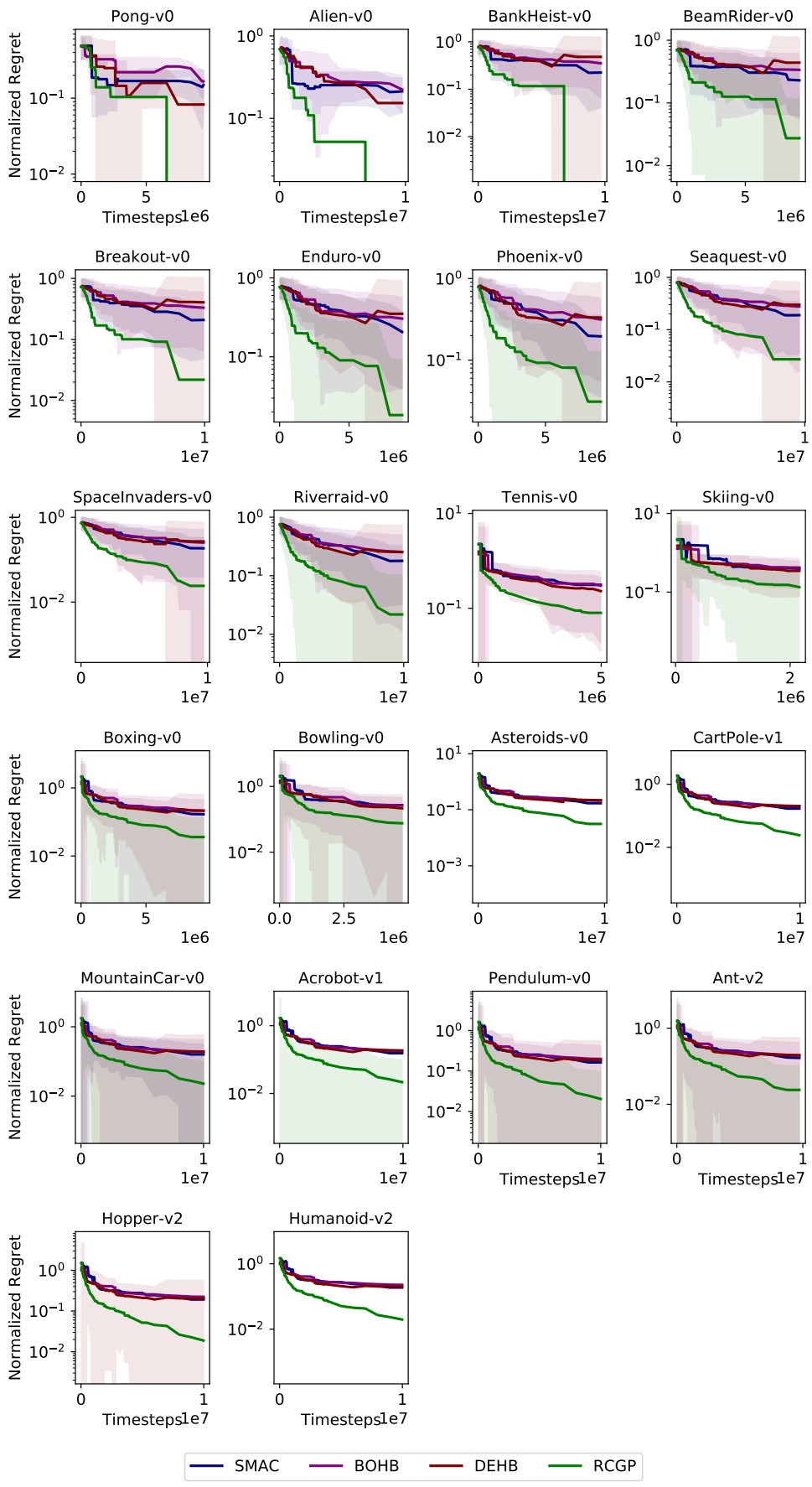

Figure 38: Performance in terms of normalized regret for SMAC, BOHB, DEHB, and RCGP in each environment for the **A2C** search space using max-smoothed episodic reward curves.

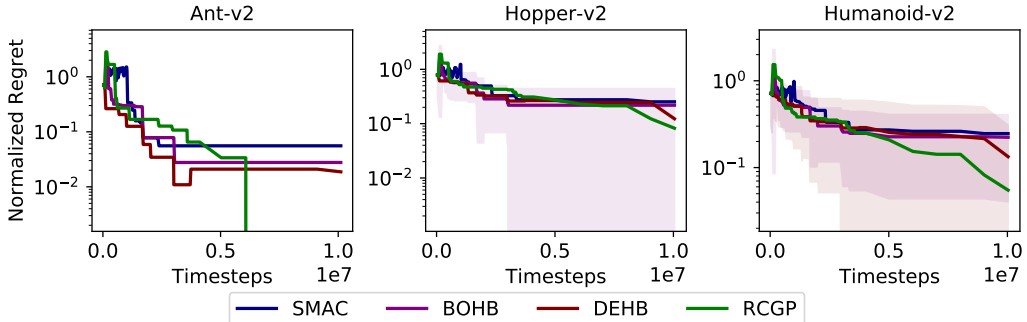

Figure 39: Performance in terms of normalized regret for SMAC, BOHB, DEHB, and RCGP in each of the MuJoCo enviroments for the **DDPG** search space using max-smoothed episodic reward curves.

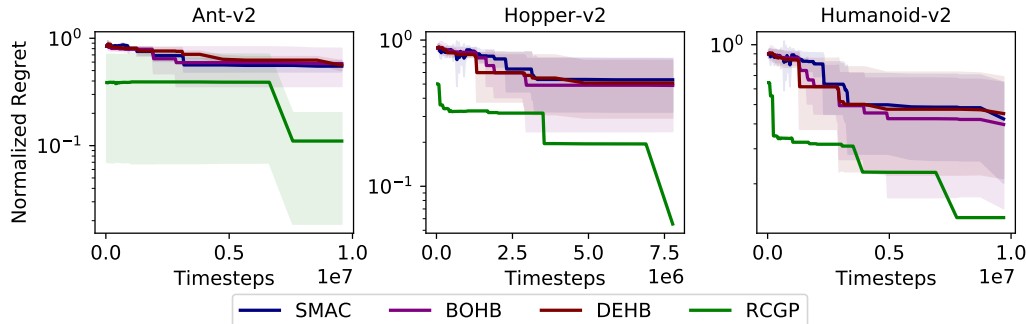

Figure 40: Performance in terms of normalized regret for SMAC, BOHB, DEHB, and RCGP in each of the MuJoCo environments for the **SAC** search space using max-smoothed episodic reward curves.

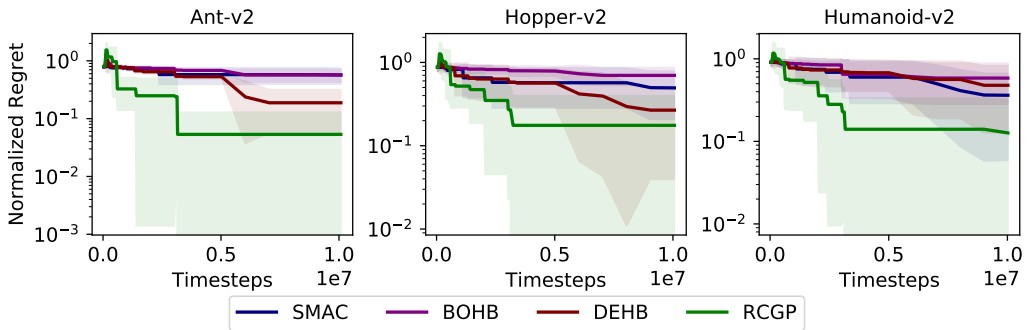

Figure 41: Performance in terms of normalized regret for SMAC, BOHB, DEHB, and RCGP in each of the MuJoCo enviroments for the **TD3** search space using max-smoothed episodic reward curves.

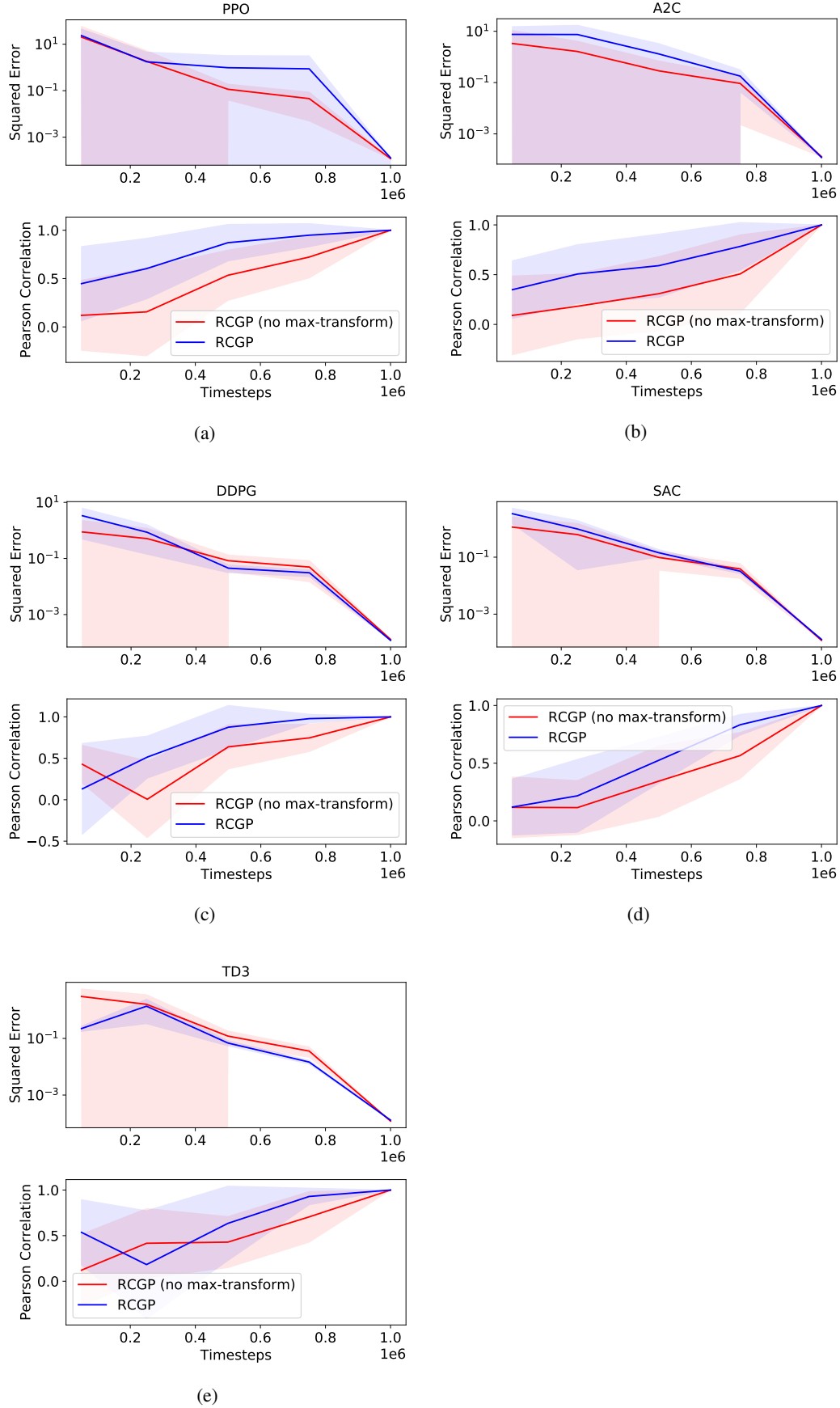

Figure 42: Predictive performance of RCGP in terms of squared error and Pearson correlation coefficient for the **PPO**, **A2C**, **DDPG**, **SAC** and, **TD3** search spaces using max-transformed episodic reward curves.

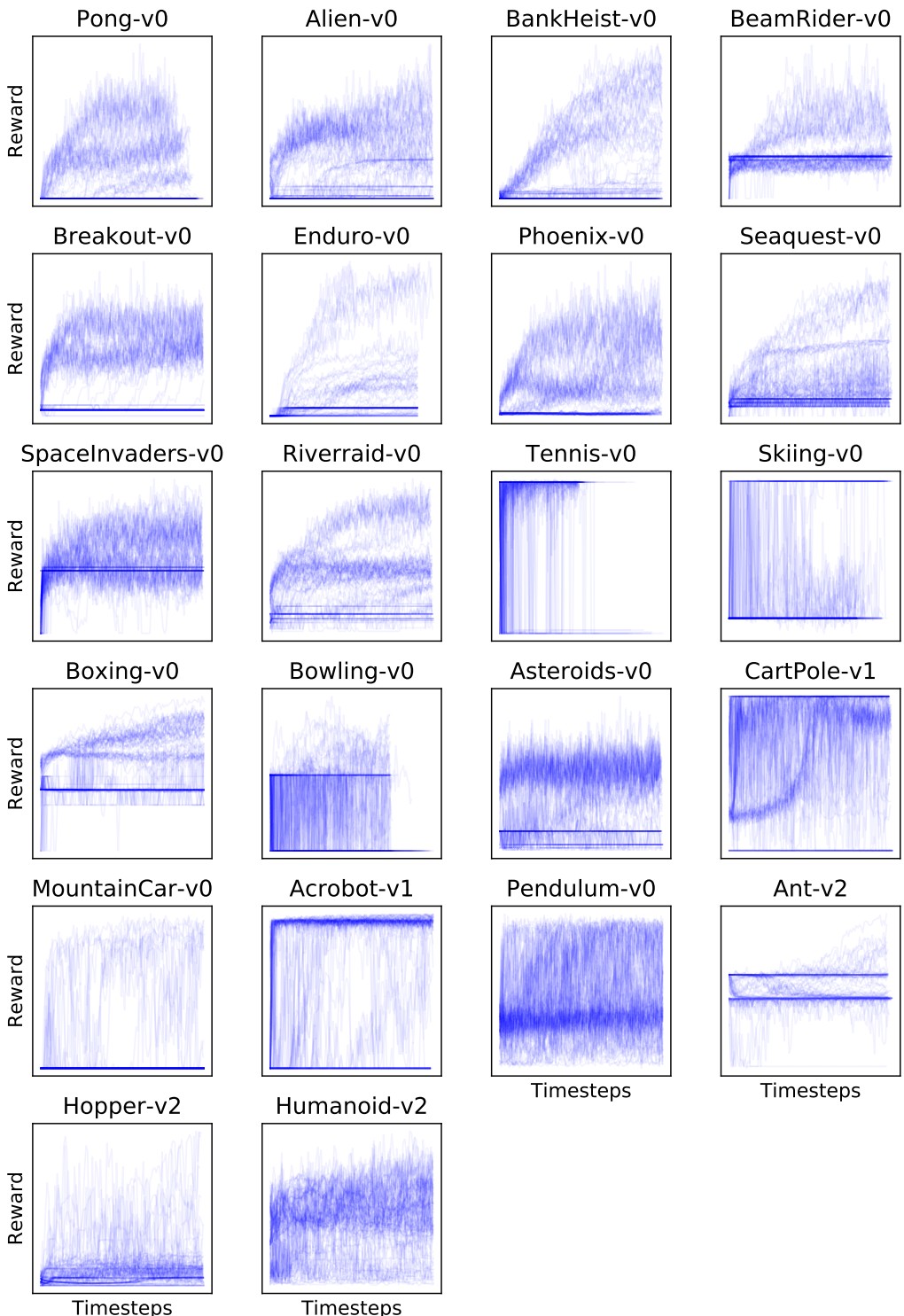

Figure 43: Reward curves of PPO on the environments included in AutoRL-Bench.

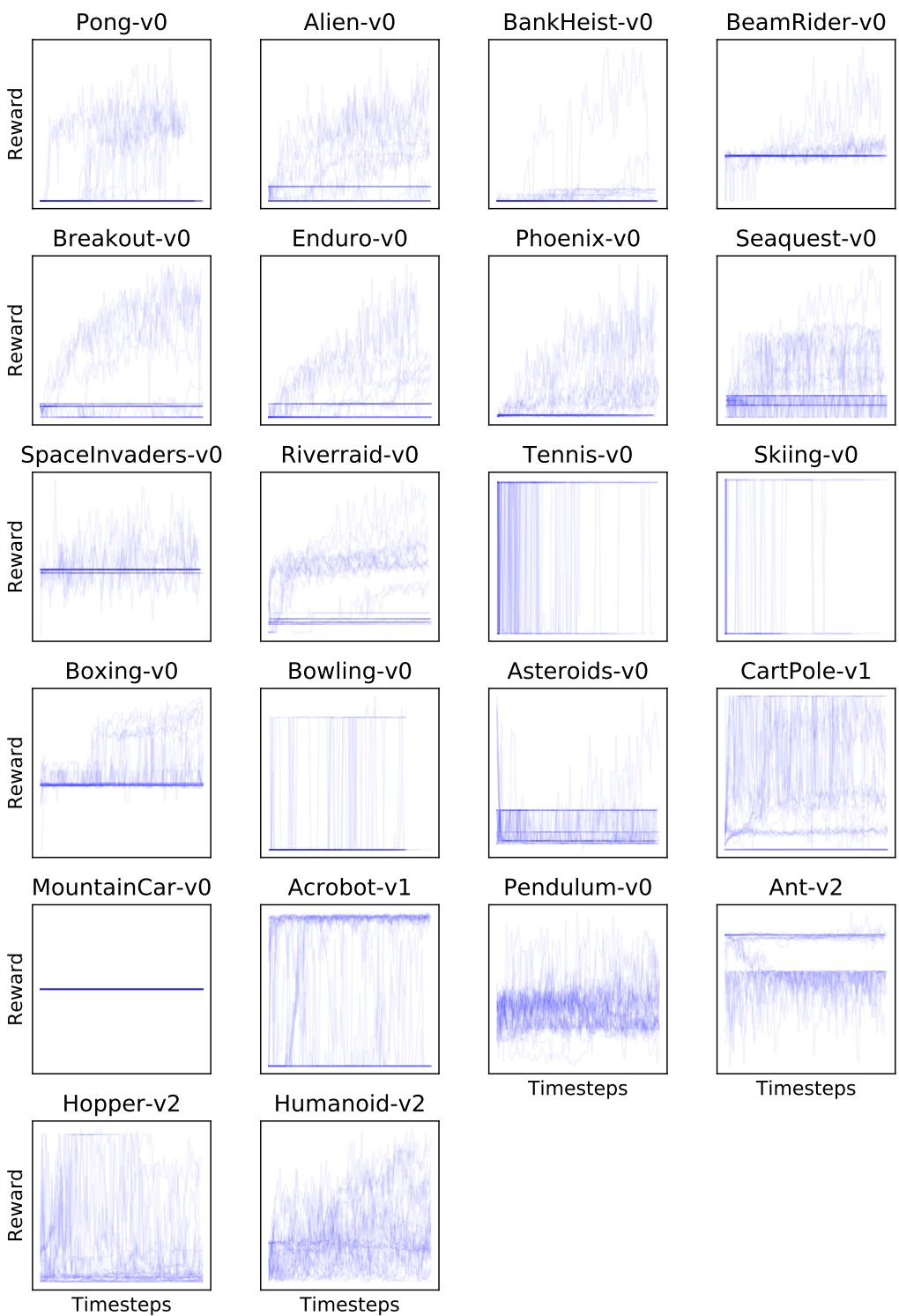

Figure 44: Reward curves of A2C on the environments included in AutoRL-Bench.

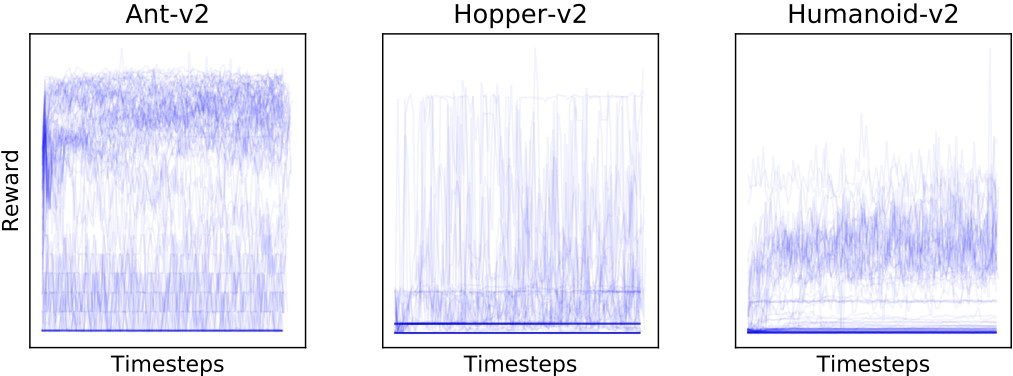

Figure 45: Reward curves of DDPG on the environments included in AutoRL-Bench.

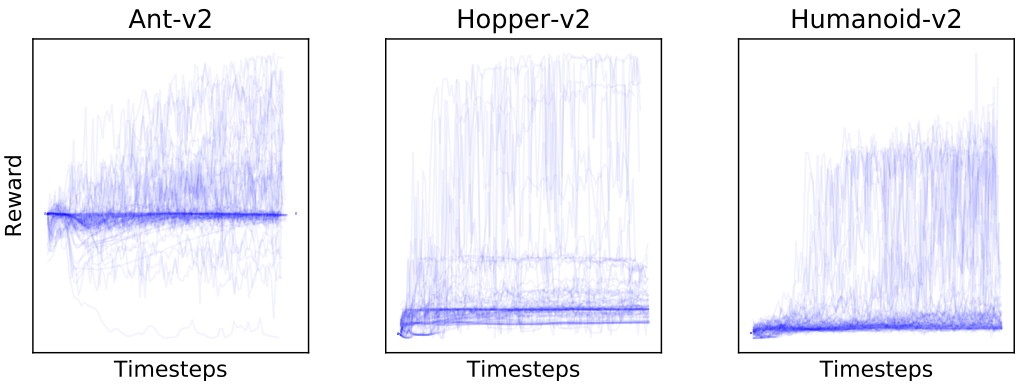

Figure 46: Reward curves of SAC on the environments included in AutoRL-Bench.

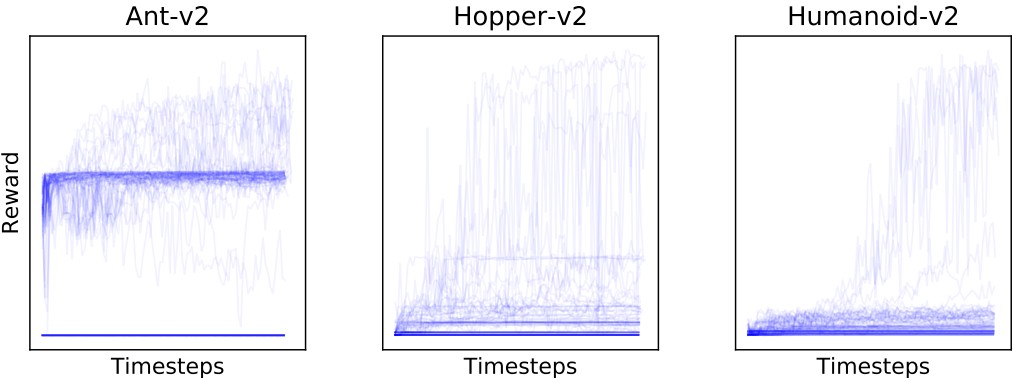

Figure 47: Reward curves of TD3 on the environments included in AutoRL-Bench.

