# OpenReview forum: "Gray-Box Gaussian Processes for Automated Reinforcement Learning"
_ICLR.cc/2023/Conference — ICLR 2023 poster_

### Official Review · Reviewer_8pAg · 2022-10-24

**Confidence:** 3
**Clarity, Quality, Novelty And Reproducibility:** The paper is mostly well-written.
**Correctness:** 3
**Technical Novelty And Significance:** 2
**Empirical Novelty And Significance:** 3
**Recommendation:** 6

**Strength And Weaknesses:**

Strengths

- The paper considers an important problem as RL algorithms are well known to be finicky with the choice of their hyper-parameters.

- The release of source code is commendable and a good contribution towards reproducibility, which is commonly poor with hyper-parameter optimization algorithms.


Questions/Suggestions for improving the paper:

- The discussion about some of the main points proposed in the paper is very limited. For example, the choice of reward model as generalized logistic function is introduced in few lines in the paper without any motivation. Please consider expanding discussion about why this is the right choice? What properties/assumption of underlying MDPs/reward model makes this an effective choice? What are the limitations of this choice? As another example, the smoothing of reward curves in Equation (1) is described in ad-hoc manner without any principle.



- Please provide more details about the training of the sigmoid coefficients and other hyper-parameters of the (multi-fidelity) Gaussian process model used in the proposed approach.

- The experimental evaluation only shows the ranking of the proposed approach and the baselines. This gives limited information about the actual performance of the proposed approach. Please show runs of individual algorithms for a particular benchmark.

- The search space consists of mainly two/three hyper-parameters which are chosen from a discrete set of values which seems too small. Can we employ the proposed approach on a continuous search space to robustly test it's effectiveness?

- The notion of gray-box has already been defined for a range of problem settings in the literature (please see [1] and references therein). Redefining the definition to `the concept of cheaper and approximate evaluations of the performance of hyperparameter configurations` makes it too specific and restricted. Please consider expanding this definition or atleast contextualizing the problem with respect to the existing (broad) literature.


[1] Astudillo, Raul, and Peter I. Frazier. "Thinking inside the box: A tutorial on grey-box Bayesian optimization." In 2021 Winter Simulation Conference (WSC), pp. 1-15. IEEE, 2021.


**Summary Of The Paper:**

The paper considers the problem of hyper-parameter optimization for reinforcement learning algorithms. A Bayesian optimization based algorithm is proposed to tackle the problem where the key idea is to model the reward curve of a candidate hyper-parameter configuration with a generalized logistic function. Gaussian Processes are used as surrogate models with an augmented input feature space consisting of the hyper-parameters, budget allocated to the configuration and the reward curve estimate. Experiments are performed on multiple RL based algorithms.

**Summary Of The Review:**

Overall, the algorithm proposed in the paper is limited in novelty and mostly an application of existing Bayesian optimization techniques for hyper-parameter optimization of reinforcement learning algorithms. I want to stress that the limited novelty alone is not necessarily a bad thing and applying existing techniques to effectively solve an important problem is quite useful. However, most of the algorithmic components employed in the paper are not motivated and lack principled justification. Please consider working on this aspect to improve the paper's quality.

---

> ### Author Response · Authors · 2022-11-19
> **Initial Response**
>
> **Concerning the choice of reward model as a generalized logistic function**: As cited in  the paper, Nguyen et al. (2020) first pointed out the sigmoidal relationship between RL reward curves and the optimization budget. Figure 1 of their paper provides example learning curves for different iterative learning settings, including two RL environments.Here we now provide a high-level intuition on why a sigmoidal shape can be expected for typical RL learning curves.
> In RL we typically start out from some random policy. So we first need some time to explore and collect data which points the RL agent in the right direction. In that early phase we can thus expect a nearly flat or only small increase in the reward curve. Once we have started improving the policy we can expect a quick increase in performance as we collect ever more helpful data which allows us to improve the current policy. In other words, in this stage we already have some successful policy which we only need to further refine, leading to quick increases in reward. Once we have found a close to optimal or very well performing policy, learning speed will slow down as any new update will not cause large deviations to the current policy, thus we will see more and more similar data points which will in the end result in very slow learning or even a plateau with no further progress.
> Besides this high-level explanation we will also provide the learning curves from AutoRL-Bench for all search spaces and environments in Figures 18-22 in the appendix.
>
> To address the noisiness of reward curves of RL algorithms we smooth the curves using a best-so-far transformation. We average rewards based on windows of h training steps, and select the highest reward at any past window (which is given in Equation 1). Thus, the max operator is only involved in how we model the reward curve and not in how we compare the performances of the individual RL algorithms. Therefore, the results of our ablation (Figures 7 and 8) show that this form of smoothing of rewards for modeling the reward curves is beneficial.
>
> **Concerning experimental details**: We provide further experimental details in the appendix, such as the baseline hyperparameters. Please note that we did follow the standard of the AutoML community and used default configurations for all baseline HPO methods as this provides a realistic comparison on how a user would employ the different HPO methods.

---

> > ### Comment · Reviewer_8pAg · 2022-11-26
> > **Reply to the author's response**
> >
> > Thank you for your detailed response. I really appreciate the effort spent in doing new experiments as well. The intuition behind logistic function seems reasonable but the figures 18-22 don't show any such recognizable pattern. The reward curves are a good addition but the proposed approach performs better than baselines in few selective benchmarks only (Figure 13). Having said that, I would like to increase my score towards accept hoping that the contributions would be toned down in the paper and explicit section adding the limitations of the underlying principle and results is added.

---

### Official Review · Reviewer_eXNC · 2022-10-25

**Confidence:** 3
**Correctness:** 3
**Technical Novelty And Significance:** 3
**Empirical Novelty And Significance:** 3
**Recommendation:** 6

**Clarity, Quality, Novelty And Reproducibility:**

Clarity

Generally good, but I think a few points could be clarified:

What kernel was used in equation 7?

What do you mean by epoch? Typically 1 epoch is one pass through the data set in supervised learning. In RL the dataset keeps changing, so epoch is ill-defined.

> gray-boy Bayesian optimization

gray-box

Quality

The breadth of the experiments and amount of comparisons, as well as the inclusion of ablation studies was good. The data analysis and presentation could have been better. I was unsure of the used benchmark, and think that running at least some small additional experiments would be good. Also the max-smoothing seemed like a poor choice based on its criticism in the literature.

Novelty

I believe the idea of using the reward curve information to better predict the outcome of the experiment is novel. There are some previous works that take into account the iterative nature of these experiments, but I believe modeling the reward curve is new.

Reproducibility

The authors provide their code, so I believe it's reproducible.

**Strength And Weaknesses:**

Strengths

- The performance appeared good.
- The ablation studies showed the necessity of the different components of the algorithm.

Weaknesses

- The AutoRL-Bench has not passed peer review yet and it is unclear whether it is suitable. Looking at the set of hyperparameters, it seems that the log10 spacing of the learning rate might be a bit large making it too easy to differentiate between the performance, so it was not clear to me whether this benchmark is really appropriate. At the same time, I understand their setup allows testing many different algorithms on a wide range of environments cheaply, and this is beneficial. I think it would be good to add your own smaller scale experiments including learning the hyperparameters from scratch on some proposed domain of hyperparameters (in an interval not between a finite selection of hyperparameters).
- The method included taking the max over past rewards in a window. This has been widely criticized in previous work in RL because it biases the evaluation. For example: Deep reinforcement learning that matters (https://arxiv.org/pdf/1709.06560.pdf)
“Due to the unstable nature of many of these algorithms, simply reporting the maximum returns is typically inadequate for fair comparison” or Deep reinforcement learning at the edge of the statistical precipice (https://arxiv.org/pdf/2108.13264.pdf)
While the ablation studies showed that this "max" was useful, comparing different algorithms when using such an evaluation protocol in RL is inappropriate. I would suggest trying some other technique, such as taking the average in the window.
- The presentation of the results could be more comprehensive and better executed. Currently they only present the ranks of the algorithms. This makes it difficult for future researchers to compare to the results in this paper. It would be useful to also present absolute performance based metrics such as the reward. This makes it also more clear how much the performance differs between the methods. Moreover, it seems that the errorbars are unrealistic in the article. They seem to wide given the smoothness of the curves. Moreover, sometimes the errorbar crosses below rank 1, which should be a hard minimum. I would suggest to change the statistical analysis performed to obtain these errorbars, e.g., using a 95% percentile bootstrap may provide better errorbars.
- There was no direct analysis of the accuracy of the newly proposed model based on the reward curves. The experiments only show the BO performance, which implies that the model should have accurately predicted the rewards; however, it would be better to directly check the prediction accuracy.
- It was unclear to me why the experiments were limited to RL. Learning curves are also present in other fields of machine learning. Is the method also useful in other domains? Why not expand the experiments to other domains? I would at least have liked to see some discussion or explanation of this.

**Summary Of The Paper:**

The work proposes to incorporate the learning curve information in reinforcement learning as features in Bayesian optimization to more efficiently perform hyperparameter optimization.

Given an RL learning curve, one could fit a generalized logistic function on the curve to obtain the parameters of the curve as features that are used in a Gaussian process model. To generalize this procedure to hyperparameters where there might not already exist a learning curve, the authors propose to predict the parameters from a neural network that takes the hyperparameters as an input.

Experiments are performed on an AutoRL-Bench a benchmark for automatic reinforcement learning. This benchmark includes learning curves for a set of hyperparameter values. And it is possible to test the performance of BO algorithms by querying the learning curves for the different hyperparameters without requiring to rerun the whole RL procedure, allowing for efficient computationally inexpensive benchmarking of BO algorithm in AutoRL. Note that this benchmark appears to be still under review and does not seem to have passed peer review yet.

They experimentally compared with many different BO algorithms including random search, GP based methods, population based methods. And the new method outperformed the baselines.

They also performed ablation studies on different components of their algorithm.

**Summary Of The Review:**

The work includes comprehensive experiments on BO in RL showing that the newly proposed method of modeling the learning curves improves the performance. However, I am unsure that the AutoRL-Bench that they performed their experiments on is a suitable way to evaluate BO algorithms as it does not appear to have passed peer review. Moreover, the analysis of the data and presentation of the results could be improved, e.g., improving the errorbars or also displaying reward information. Taking the max of the rewards as done in this paper has also been criticized in prior research. Considering these points, I am leaning towards suggesting to reject.

Update
------------------------------
_________________

After seeing additional details in the updated paper, and more discussion with the authors, I perceive more issues in the paper than initially, and I have reduced my score to 3. The main additional issues are that I believe the performance of the algorithms should also be compared based on the performance after fully training with the currently selected best hyperparameter (as opposed to looking at the reward at the current budget), and that the performance on several of the environments is quite poor (as can be seen from the reward curves). Regarding the second issue, it would have been good to include results for larger hyperparameter optimization budgets (the table based benchmark allows quickly testing large budgets, so I guess there is no computational issue with including larger budgets).

I list my main concerns below:
- Only indirect evidence is provided that the predictive model is performing well. I pointed this out in my initial review by saying that the accuracy of the predictive model was not compared. The authors responded that the accuracy is not that important, and the ranking is more important. While this may be true, it does not address the intention of my comment, and if the ranking is more important, the work could have instead investigated the ranking accuracy. Other ways to provide direct evidence that the predictive model is performing well would have been to visualize the predictive results. Providing this kind of evidence is important to prove that the algorithm is working as intended. There could be any number of reasons why the performance of the algorithm appeared to be good (e.g., bugs in the code or inappropriate evaluation), and providing multi-faceted evidence is a good way to mitigate such concerns. It may be particularly good if the authors compare with a simple ranking method, e.g., ranking the hyperparameters based on the reward at the same budget, and compare whether the predictive model outperforms such a simple method.
- The benchmark is not well established. In my original review, I said that the benchmark has not yet been published. The authors responded that after the reviewing started the paper was accepted for publication at a NeurIPS workshop. This does not mitigate the issue that the benchmark is not yet established in the community as a good way to evaluate algorithms as there are no prior works using this benchmark. Workshop papers are often work in progress that still have issues. The authors also asked whether I could provide any other benchmark for them to use. Actually, I don't necessarily think that the benchmark is bad, and it seems like a really good idea to have such a tabular benchmark for HPO; however, as it is very new, I am not convinced of its reliability. If the benchmark were well established, the requirement for additional evidence in the previous concern could be lessened; however, as the benchmark is not well established, I suggest that the authors should mitigate such a concern by providing more thorough experimental evidence, e.g., also include an evaluation of the predictive model's performance.
- Some missing details. I couldn't find reward curves for the BOHB, SMAC, DEHB methods. Moreover, the reward curves for the proposed method but without the max smoothing were also missing. The reward scale on the axes in Figures 18-22 was also missing (making it difficult to see whether the performance of algorithms is actually good, and also difficult to see whether the HPO found good configurations). Also, it was not written what kernel was used in equation 7 (the authors later clarified this for me, but not in the initial response, and the paper was also not updated).
- The performance of all algorithms is poor on several environments (see figures 13 and 14). I believe the HPO optimization budget should be increased, so that it finds a good performance. The authors emphasized that their method works well for low budgets, but I don't see any good reason why they should withhold the results for high budgets. It would be important to know whether the new method is better for both low and high budgets, or whether it is only better in the early stages of HPO. Due to the use of the tabular benchmark that does not actually require running the RL algorithms, the computational speed of the experiments should be fast, and I don't see an issue with experimenting with higher budgets.
- **Evaluation** The exact method of evaluation is still a bit unclear to me due to miscommunication between me and the authors. But my current understanding is the following: There is a total maximum budget $B$, and each hyperparameter setting $i$ has their total allocated budget until the current step $b_i$. The database contains learning curves for each hyperparameter setting for $T > b_i$ steps. The algorithms work by selecting one configuration $i$, incrementing its budget to $b_i + \Delta b$ and querying the database for the performance at the budget $b_i + \Delta b$. This performance is stored in the history. I believe that these parts I was able to confirm from the code. How this history is finally used for evaluation, I am a bit unsure about, as the authors did not provide the code they used for postprocessing and plotting. I believe they take the max in this history, and take that as the evaluation of the algorithm.

I perceive two issues with this evaluation method. First, the evaluation is only based on the performance during HPO with a policy that is not fully trained. In HPO we could consider two different ways of evaluating: evaluate the policy at the current budget, or evaluate the policy after fully training with the chosen hyperparameter. I believe that the current method of evaluating only the policy at the current budget is insufficient, as often we may be interested in the performance of the chosen hyperparameter rather than the currently best trained policy. For example, if a HPO method increments all hyperparameters by only a small fraction of $T$, and selects the best hyperparameter from there, the selection performance may be quite good, while because the policy is only trained for a little bit, the performance at the current budget will be poor. For this reason, I believe that the evaluation should at least include the final performance as well (perhaps providing both methods of evaluation would be good).

**Max** I had quite a bit of miscommunication with the authors about the max smoothing. My initial comment was about max smoothing during the budget increment in the interval $[b_i, b_i + \Delta b]$, and the authors responded that this does not affect the evaluation of the algorithm. This part I mostly agree with (although I believe that other comparisons with other smoothing methods would also be good, as I am skeptical of this method). However, my later comment is not about this smoothing. If my above explanation of the algorithm is correct, the history contains the final performance of each budget increment. This means that there will be multiple entries for each learning curve (if a hyperparameter is incremented multiple times). I believe for their evaluation, they pick the maximum value in the history, so essentially, it will be taking a max over multiple points from each learning curve. This would add a selection bias and overestimate the performance (as the method would select the point where the randomness made the reward be high). This would be particularly problematic when the increment size is different for the different HPO algorithms (e.g., in the comparison with the single fidelity methods). This explanation matches what is described in the algorithm 1, where the max is taken in line 9 (note that algorithm 1 is titled "Gray-Box HPO for RL" so I believe it corresponds to all their HPO algorithms). If the authors wish to include the performance comparison during training (i.e., not only after fully training as I suggested above), I believe they could mitigate this issue by comparing the performance at the highest budget for each hyperparameter (i.e., not the max in the history including all increments).

- The performance for the other algorithms is not provided for the max smoothing, so it remains unclear whether the new method performs well because of the max smoothing or because of the reward modeling. Here the authors did provide an ablation study showing that the max smoothing improved their performance. Moreover, in their rebuttal they argued that it is nonstandard to immediately use the new method for all previous algorithms as well. However, in this case it would at least be necessary to know how the reward modeling without the max smoothing compares to the previous algorithms. At the moment this is impossible to know, as the only results without max smoothing are rank comparisons in an ablation against the standard reward curve method. The work should at least have included reward curves for the method without max smoothing.

- Data processing and presentation could be better, e.g., more realistic errorbars.

Update 2
------------------------------
_________________

The authors performed substantial additional experiments, and provided a detailed new response, so I have increased my score to 6 and would be happy to see the work published. I refrained from setting the score higher because it is quite late in the review process, and it is no longer possible to update the paper, but I guess the changes for the camera ready will not be difficult if the paper is accepted.

Regarding the author response: they performed experiments looking at the predictive accuracy of the models, which also showed that the max smoothing improved the ranking accuracy while reducing the prediction accuracy (providing more evidence for the authors' intuition about why max smoothing may be useful). This could be further improved by showing a few visualizations of the predicted reward curves.

They also provided the missing additional reward curves. Moreover, they explained that the performance is close to optimal on most environments (after looking at the plots on Fig 13 and 14 again, I guess the performance seems poor because the scale on the axis is inappropriate for some environments to properly see the performance).

The authors now also provided a clear explanation of their evaluation protocol as well as added the code for the post processing from where I was able to confirm the evaluation method, and it was fine.

They also performed new experiments trying out max-smoothing also with the other methods.

One remaining small issue in the paper are the errorbars. After looking at the code, it seems the errorbars are the standard deviation instead of the standard error (actually this was also written in 1 place in the paper), so I believe it should be simple to fix, and I'll give a suggestion in a comment below.

---

> ### Author Response · Authors · 2022-11-19
> **Initial Response**
>
> **Concerning AutoRL-Bench**: According to the [repository of AutoRL-Bench](https://github.com/releaunifreiburg/AutoRL-Bench.git), the benchmark was accepted at the Meta-Learning workshop that is peer-reviewed and will soon be held as part of NeurIPS and offers expertise in AutoML.
>
> **Concerning the max operator**: To address the noisieness of reward curves of RL algorithms we smooth the curves using a best-so-far transformation. We average rewards based on windows of h training steps, and select the highest reward at any past window (which is given in Equation 1). Thus, the max operator is only involved in how we model the reward curve and not in how we compare the performances of the individual RL algorithms. Therefore, the results of our ablation (Figures 7 and 8) show that this form of smoothing of rewards for modeling the reward curves is beneficial.
>
> **Concerning presentation of results**: We provide new plots showing the obtained rewards in the appendix. Please see Figures 13-17 showing the average evaluation rewards for the suggested configurations for RCGP, PBT, and PB2 for each search space and each environment.
>
> **Concerning the accuracy of the proposed model**: The job of our reward curve model is to inform the HPO which configuration at a lower fidelity is worth continuing running. Thus its objective is not to accurately predict reward values but rather provide a reasonable ranking of which configurations can be expected to result in good rewards. We argue that the choice of the max operator in smoothing over the noisiness of RL reward curves will result in our model giving overestimating predictions. However, our ablation shows that this is actually desirable and is especially helpful in the initial phases. Comparing this to RCGP without the max smoothing, which can be expected to make more accurate predictions, we can see that in the long run not using max-smoothing does not provide a clear benefit.
>
> **Concerning other target applications**: It is true that in other applications one could use reward curve information. However, from our perspective, RL is a crucial domain which has the promise of providing solutions to a broad field of applications. Still, applying RL out-of-the-box is neigh on impossible without extensive expert knowledge. To the best of our knowledge no dedicated HPO methods exist for RL that prioritize sample-efficiency. To enable the broader research community to make use of the power of RL we deem it important to provide methods that explore how to get the best out of RL agents while wasting as few resources as possible, which we did by incorporating reward curve information.

---

> > ### Comment · Reviewer_eXNC · 2022-11-21
> > **Thank you for the response**
> >
> > **Regarding the max** If my understanding is correct, the algorithms are also evaluated by taking the maximum performance along the training curve (bottom of Algorithm 1). In this sense, the max operator in your reward curve model, and in the way you evaluate the algorithms matches. However, I believe that taking the maximum along the reward curve is a biased estimate, and it seems inappropriate to evaluate the RL algorithms based on this metric. Using the maximum along the reward curve would be OK if the algorithms are evaluated with a large number samples at each training step, so that the estimates are accurate; however, this appears not have been done. And taking the max would overestimate the performance at the selected training step, so it seems inappropriate. It would help if the benchmark includes separate evaluation episodes after x amount of training episodes, evaluated accurately with many samples and not included in the training data.
> >
> > I also found it suspicious that in the newly posted reward curves, the performance of RCGP increases monotonically (implying that it is evaluated using its historical max) while the other methods have an oscillating performance.
> >
> > Several of my comments were also not addressed (requests for clarifications, etc.), so I feel inclined to keep my score.
> >
> > A HPO optimizer like this would be useful to the community, so I encourage the authors to continue working on the experimental analysis and presentation of the results.

---

> > > ### Author Response · Authors · 2022-11-21
> > > **Continued Discussion**
> > >
> > > We believe there is a misunderstanding at this point.
> > > A HPO method for is composed of two stages (i) search and (ii) evaluation.
> > > After step (i) has completed, the best discovered hyperparameter configuration is evaluated using the experimental protocol in step (ii).
> > >
> > > Each HPO method has its own search strategy, including its assumptions and design choices. For instance, our search strategy is based on exploiting the reward curve information, PBT/PB2 rely on evolutionary search, etc. A HPO method is entirely free to make its own design assumptions, as long as it does not violate search constraints, hyperparameter space definition, environment definition, and, the wallclock budget plus memory constraint. What matters is the quality of the discovered hyperparameters, found within the same budget and constraints.
> > >
> > > However, there is ONLY one evaluation protocol for all methods (step (ii) must be universal and same for all methods), which evaluates the performance of the best discovered hyperparameter configurations of rival methods.
> > >
> > > In that context, Algorithm 1 describes our search strategy for the hyperparameters of an RL method on a specific environment, given a specific time budget. Algorithm 1 is step (i) of our method, and not the evaluation protocol (step (ii)) that describes how methods are evaluated.
> > >
> > > The evaluation protocol for all methods is: "We evaluated the trained agent with the best hyperparameter configurations for 10 episodes and report the mean final reward of these episodes. I.e. The rank performance plots shown for a x-axis time budget B mean:
> > > -Search for the best hyperparameter configuration for a budget of B time.
> > > -Evaluate the best configuration (found with a search budget of B) for 10 episodes and measure the mean final reward.
> > > -Rank methods based on the mean final reward".
> > >
> > > **We double stress that the max smoothing is an assumption of our technique's search procedure (step (i)) and is NOT used for evaluating the best configurations of the rival methods (step (ii)).**
> > >
> > > Furthermore, please find the implementation of our evaluation protocol in file https://anonymous.4open.science/r/RCGP-65CC/source/lcgp_arlbench.py line 326 of our source code. “results["eval_avg_returns"][budget -1]” requests the average evaluation reward at the highest budget (i.e. fully trained RL agents). Our utils.py file (https://anonymous.4open.science/r/RCGP-65CC/autorlbench_utils.py) simply handles queries to AutoRL-Bench and maps the keyword “returns_eval” to “eval_avg_returns”. Looking at appendix B of AutoRL-Bench (see https://openreview.net/pdf?id=RyAl60VhTcG) you can see that returns_eval is **the evaluation reward and that this is the average over multiple rollout episodes and not the maximum reward**.
> > >
> > > As a result, we hope to have cleared the confusion. It would be great if you revise the score considering that the criticism is based on a misunderstanding.
> > >
> > > **Addressing Clarification questions**
> > >
> > > Please excuse that we overlooked some clarifications. We answer them in the following here:
> > >
> > > **What kernel was used in equation 7?**: We used a Matern 5/2 kernel for the GP.
> > >
> > > **What do you mean by epoch?**: We agree that for the case of RL, an epoch is ill-defined. As the gray-box hyperparameter optimization literature mostly includes methods for AutoML, we were describing how these methods are generally employed in supervised learning. In the case of deep RL and the specific case of our experiments, one unit of budget equals training an agent for 104 timesteps in the training environment, then evaluating it in the evaluation environment for 10 episodes and reporting the mean.
> > >
> > > We would be glad to answer any pending point you believe we might not have clarified fully.

---

> > > > ### Comment · Reviewer_eXNC · 2022-11-22
> > > > **More clarification**
> > > >
> > > > I am still a bit confused by your explanation. Do you evaluate the best policy that was found within the budget B? Or do you query the final training performance of the best hyperparameter (i.e., as I understood, if one full training run takes T steps, then your budget is B=10T; is the evaluation based on the reward after training for T steps)? Also, I guess the evaluation should use 10 different training runs (each with different seeds), and each one evaluated for multiple episodes at the end of training (if each episode is stochastic).
> > > > If I understand you correctly, for the evaluation, at each step you query the final performance of the currently selected best hyperparameter (i.e., not the performance during training), and plot the ranking and reward curves based on this final performance? In that case, it's fine. But I think it's not very clear in the paper. Moreover, I still think that it would be interesting to compare to other smoothing methods, e.g., using the mean. Taking the max is typically not a good evaluation method. While the results showed that it improved the optimization performance by smoothing the reward, it's effectiveness could be better justified by comparing to other methods that smooth the reward, e.g., comparing with averaging. Also, were the reward curves for the other methods also max-smoothed?
> > > >
> > > > Please note that the Openreview page you linked to with the AutoRL-Bench has restricted access. I guess only the authors and reviewers can see it.
> > > >
> > > > I feel I won't be increasing my score. Even without the issue with the max, I still perceive the following weaknesses:
> > > > clarity, lack of evaluation of the accuracy of the predictive model (this was in my review but was not addressed), the errorbars seem unrealistic, the benchmark was only recently accepted for publication and there is still no accessible documentation for it, no experiments on continuous hyperparameter optimization domains.

---

> > > > > ### Author Response · Authors · 2022-11-22
> > > > > **Continued Clarification (part 1)**
> > > > >
> > > > > Your understanding of the evaluation is correct. The policy belonging to the best hyperparameter configuration for any budget is evaluated for 10 episodes and the mean final reward is reported as the comparison metric. These experiments are repeated 10 times with 10 different seeds.
> > > > >
> > > > > Although we clarify our protocol in Section 4.1, we will make the evaluation part longer and more detailed for the camera ready.
> > > > > In your answer above "Thank you for the response" dated 21.11.2022 you mention that the main criticism is the usage of max smoothing for evaluating configurations.
> > > > > That expressed criticism does not hold, as both sides converged on the fact that we use the standard protocol of evaluating hyperparameter configurations in RL (repeat for several seeds and measure the mean final reward of multiple rollouts).
> > > > >
> > > > > Coming to the points of your latest follow-up post "More clarification" dated 22.11, you commented on a couple of points:
> > > > >
> > > > > **Regarding the usage of max-smoothing for the other methods**:
> > > > >
> > > > > This is valid for only a subset of methods that use the learning curve information, such as  BOHB, DEHB, and SMAC, not RS, GP, PBT, and PB2. We did not deploy max-smoothing to BOHB, DEHB, and SMAC, because that is not a feature of these methods, but a recent hypothesis and a contribution of our current paper. We hypothesized that the max-smoothing of reward curves gives a performance lift for our method, which we then defended empirically. We believe it is unorthodox to propose a new hypothesis and immediately deploy it to all previous baselines within the same paper. Once this paper is accepted, we hope it motivates a stream of research on using smoothing strategies for reward curves in the realm of gray-box HPO. Then the community and the authors of other HPO methods can adopt these transformations in their future works.
> > > > >
> > > > > **On the AutoRL benchmark**:
> > > > >
> > > > > In your original review, you raised a weakness in our work, stating that "AutoRL-Bench has not passed peer review yet and it is unclear whether it is suitable". We replied by clarifying that the work has already passed peer reviewing. In your recent response, you revised your criticism to "the benchmark was only recently accepted".
> > > > >
> > > > > We believe this criticism point does not hold anymore, for two reasons: i) the benchmark is indeed peer-reviewed at a NeurIPS 2022 workshop, and ii) it is the ONLY public benchmark with precomputed reward curves for RL.
> > > > > Re-running all experiments from scratch for 5 RL methods (DDPG, A2C, PPO, SAC, TD3), 22 environments (OpenAI Gym: Mujoco, Atari, Classic Control), and 7 HPO baseline is computationally-challenging without a tabular benchmark.
> > > > >
> > > > > In that context, no other benchmark exists that is more well-established, or older. If the reviewer is aware of an alternative, we are glad to consider it.
> > > > >
> > > > > Regarding the PDF of the paper, the NeurIPS workshop has just recently disabled access to the review pages of the papers at OpenReview. However, the pdf is also accessible from the GitHub of the benchmark's page at this [link]( https://github.com/releaunifreiburg/AutoRL-Bench/blob/main/AutoRL_Benchmark.pdf).

---

> > > > > > ### Author Response · Authors · 2022-11-22
> > > > > > **Continued Clarification (part 2)**
> > > > > >
> > > > > > **On the accuracy of the prediction model**:
> > > > > > As we clarified in our original response, although we could run experiments comparing the forecasting quality in estimating the reward curves, we did not see a rationale behind it, because forecasting is not the task we are addressing. Our task is hyperparameter optimization, which is not equivalent to reward curve forecasting. Forecasting optimizes for reducing the residual in estimating the final performance, while HPO optimizes the top-1 validation accuracy of the best hyperparameter configuration. The aim in HPO is the top-1 metric, not the forecasting one.
> > > > > >
> > > > > > Consider the following example:
> > > > > >
> > > > > > - learning curve 1 is 4 points long 0.1, 0.2, 0.4, 0.5
> > > > > > - learning curve 2 is 4 points long 0.2, 0.4, 0.45, 0.6
> > > > > >
> > > > > > assume only the first two points of each curve are observed.
> > > > > >
> > > > > > consider two hypothetical methods that estimate the final (fourth) point at each curve using the first two:
> > > > > >
> > > > > > method 1:
> > > > > > - learning curve 1: observed: 0.1, 0.2, real last value: 0.5, estimated last value: 0.8
> > > > > > - learning curve 2: observed: 0.2, 0.4, real last value: 0.6, estimated last value: 0.9
> > > > > >
> > > > > > method 2:
> > > > > > - learning curve 1: observed: 0.1, 0.2, real last value: 0.5, estimated last value: 0.55
> > > > > > - learning curve 2: observed: 0.2, 0.4, real last value: 0.6, estimated last value: 0.50
> > > > > >
> > > > > > It is obvious that method 2 is better at forecasting based on the approximation accuracy, however, method 1 is better at HPO because the rank of the configurations is preserved. Therefore, regression quality is not the metric that we care about in HPO.
> > > > > >
> > > > > > We already clarified this point in our answer "Initial Response" dated 19.11 as "... objective is not to accurately predict reward values but rather provide a reasonable ranking of which configurations can be expected to result in good reward ...".
> > > > > >
> > > > > > **On the lack of clarity**:
> > > > > > In this regard, the other three reviewers have diverging opinions
> > > > > > - Reviewer ETq6: "High clarity: Well written with excellent related work context. ..."
> > > > > > - Reviewer h99m: "The paper is mostly well-written. ..."
> > > > > > - Reviewer 8pAg: "The paper is mostly well-written."
> > > > > >
> > > > > > In any case, we respect your comments on the clarity and we will do our best to revise the manuscript for the camera ready. Could you be a bit more specific regarding the sections that you think need more clarity?
> > > > > >
> > > > > > In summary, we do not perceive open technical points from your comments, and we believe to have clarified all concerns. We would be glad if you could consider re-scoring our work in light of our clarifications.

---

> > > > > > ### Comment · Reviewer_eXNC · 2022-11-22
> > > > > > **More clarification**
> > > > > >
> > > > > > Still, I find the explanation to be imprecise. Do you mean 10 episodes or 10 training runs?
> > > > > > By mean final reward, do you mean the mean final return?
> > > > > > By policy belonging to the best hyperparameter configuration do you mean the policy after fully training with the best hyperparameter configuration?
> > > > > >
> > > > > > Regarding the evaluation protocol, I don't see it explained anywhere in the paper. It is further confusing because there is both an evaluation of which hyperparameters to continue training on, and the final evaluation of the hyperparameter. Probably the clarity can be fixed, but I think this is an issue in the current draft.
> > > > > >
> > > > > > Regarding using max-smoothing for the other methods. If you don't use max-smoothing for the other methods it remains unclear whether your method improved because of modeling the learning curve or because of max-smoothing. If you don't want to implement max-smoothing for the other methods, one way is to compare your algorithm without max-smoothing to the other methods, then show that by using max-smoothing you can further improve your method.
> > > > > >
> > > > > > Did you include the reward curves for BOHB, SMAC, DEHB, and RCGP somewhere?
> > > > > >
> > > > > > On the RL Benchmark: the paper was added to github less than 2h ago. I also believe the benchmark was not published at the time of submitting the article, and I was not able to find any documentation for it earlier either. I think this documentation should have been available for the initial review. The update to the github page about the paper having been accepted was also added only a few days ago.
> > > > > >
> > > > > > Regarding the prediction accuracy of the model: I am interested in the values, because for example if the predictions are very wrong, it would imply that there is something strange in the algorithm. Checking the prediction results acts as a sanity check. It would be useful to know whether the algorithm is working well because the predictions are accurate, or despite the predictions not being accurate. This would also be useful information for others trying to replicate your work. Providing more data in your paper is always helpful. Moreover, I am not convinced by your argument of the rankings being more important, because your prediction model does not include any special component to make it focus on the rankings. I rechecked your initial response about the prediction accuracy, but I understood that section to be discussing about the max operator, not about why you didn't include prediction accuracy results.
> > > > > >
> > > > > > "you mention that the main criticism is the usage of max smoothing for evaluating configurations".
> > > > > > No, I don't think I wrote that that is the main criticism. Also, in one of your earlier posts you used quotation marks for your explanation, when this was not actually quoted from your paper. I find this a bit misleading.
> > > > > >
> > > > > > In summary, I think the paper is not ready. I think it can be improved by increasing the clarity, reporting more results, more investigation of the max-smoothing (whether average smoothing would also work, whether the proposed method still works better than the other algorithms without max-smoothing), better confidence intervals, investigating the prediction accuracy of the model and discussing the results, perhaps adding smaller experiments on a continuous domain.

---

> > > > > > > ### Comment · Reviewer_eXNC · 2022-11-26
> > > > > > > **I took another look**
> > > > > > >
> > > > > > > After taking another look at the code, etc.
> > > > > > >
> > > > > > > I guess the evaluation is not based on the final return after fully training with the chosen hyperparameter, but on the policy at the current budget.
> > > > > > >
> > > > > > > One could envision both evaluation methods: the return at the currently trained policy, or the return after fully training with the currently chosen best hyperparameter. However, I think that the evaluation after fully training should at least definitely be included. The main task of HPO is to select the best hyperparameter. Some HPO method may make the selection based on only training each hyperparameter for a fraction of the full training run. To fairly compare with such methods, a comparison based on the final return should be included I think.
> > > > > > >
> > > > > > > Some other technical issues I perceive are:
> > > > > > > If we look at the reward curves, on many of the tasks, none of the algorithms find a good solution. I believe results with larger optimization budgets should also be included, e.g. see Fig 13, 14.
> > > > > > >
> > > > > > > Some of the reward curves were missing.
> > > > > > >
> > > > > > > One of the other reviewers also mentioned this, but RS and GP query each hyperparameter at the max budget, allowing them to only make 6 additional choices. Comparing with a method that queries for a shorter optimization length would be useful as often it is not necessary to run the full training procedure to pick a good hyperparameter.
> > > > > > >
> > > > > > > Repeating my previous point about the accuracy of the predictive model: while it is not the direct metric you are interested in, it serves as a sanity check. Another way may be to show visualizations of the predicted curves and how they compare to the real curves. If the argument was that the ranking is more important than the accuracy, one could compare the rankings based on the predictive model. The main purpose of providing such data is to prove that the algorithm is working as intended. Currently, only indirect evidence was provided that the predictive model is working well. For example, if the predictive model currently just outputs 0 at each argument that would make no sense, and I wouldn't want to recommend the paper for publication. There was no direct evidence that the model is not outputting such strange values.

---

> > > > > > > > ### Comment · Reviewer_eXNC · 2022-11-26
> > > > > > > > **One extra question**
> > > > > > > >
> > > > > > > > In the code, in the data stored on line 332, when you use this data for the final evaluation, do you take the max return in this data? This data includes multiple data points from each training curve, so that would be adding in the max bias as I was explaining earlier (as it takes the max along the included points in the training curve rather than taking the final value).
> > > > > > > > Moreover, my current understanding after reading the code is that Algorithm 1 is indeed how you evaluate the performance of your algorithms. where Line 7 in the algorithm corresponds to line 332 in the code.

---

> ### Comment · Reviewer_eXNC · 2022-11-29
> **Update: reduced score**
>
> I'm afraid I have reduced my score to 3. The details can be seen in the Update subsection in the Summary section of my main review. If the authors have any comments, please provide them in this thread. In particular, I am still not completely sure of the evaluation method of the algorithms, and it would be good to have a definitive clarification regarding this point.

---

> > ### Author Response · Authors · 2022-12-01
> > **Demanded experiments (1/2)**
> >
> > As a response to the reviewer’s concerns, we have run a battery of new experiments, concretely:
> >
> > - Experiments to assess the predictive accuracy of the surrogate
> > - Experiments with variants of the multi-fidelity HPO baselines using max-smoothed reward curves
> > - Experiments of the gain in running the HPO process for longer budgets.
> >
> > **On the accuracy of the predictive model**:
> >
> > Following the reviewer's comment on observing the predictive accuracy as a potential debugging step of the algorithm, we ran additional experiments for measuring the predictive performance, and the rank correlation of our GP surrogate.
> >
> > The plots below show as y-axis (i) the predictive error (least square error) and (ii) the Pearson’s correlation. Our GP is trained with the partially-observed reward curves (until the point indicated in the x-axis), to estimate the final return. The ground-truth final return is always the non-max-smoothed original reward curve.
> >
> > The results for all five methods are at https://anonymous.4open.science/r/RCGP-65CC under the folder “RCGP Predictive Performance Plots”. For each plot, we show the variance across all the environments.
> >
> >
> > The results highlight the following findings:
> >
> > - In terms of the predictive accuracy (least-square error) on forecasting the final return of the original raw (non-max-smoothed) reward curve:
> >      - The forecasting error decreases as a longer segment of the reward curve is observed, validating the correctness of our GP fitting procedure.
> >      - The forecasting error of the GP variant that uses the max-smoothed curves for training is higher than the GP that directly uses the raw reward curves. This is also a logical finding, because the ground-truth final return is always the non-max-smoothed version of the curve, therefore, the GP that learns using the same non-transformed curves as the ground-truth has a smaller predictive error.
> > - However, in terms of the correlation of the ranks of the predicted vs. the ground truth values of the final return:
> >      - The GP that is fitted from max-smoothed curves has a significantly larger correlation between estimated and final return values.
> >      - As the correlation is more essential for HPO than the forecasting loss, these experiments provide a strong analysis explaining the superiority of our method in terms of the HPO performance.
> >
> >
> > **On the benchmark**:
> >
> > We thank the reviewer for supporting the idea of using a tabular benchmark, and we agree with the remark: "the authors should mitigate such a concern by providing more thorough experimental evidence, e.g., also include an evaluation of the predictive model's performance.".
> >
> > We believe that the presented experiments on the predictive performance address the concern.
> >
> > **On comparing to the max-smooth version of the multi-fidelity baselines BOHB, DEHB, SMAC**:
> >
> > We conducted new experiments where the baselines BOHB, DEHB, and SMAC also use max-smoothed reward curves during HPO. Our method clearly outperforms the baselines in terms of the quality of the discovered hyperparameter configuration.
> >
> > The additional results are shown at:
> >
> > https://anonymous.4open.science/r/RCGP-65CC/REWARD_TD3_log_reward_all_mf_amax.pdf
> >
> > https://anonymous.4open.science/r/RCGP-65CC/REWARD_SAC_log_reward_all_mf_amax.pdf
> >
> > https://anonymous.4open.science/r/RCGP-65CC/REWARD_PPO_log_reward_all_mf_amax.pdf
> >
> > https://anonymous.4open.science/r/RCGP-65CC/REWARD_DDPG_log_reward_all_mf_amax.pdf
> >
> > https://anonymous.4open.science/r/RCGP-65CC/REWARD_A2C_log_reward_all_mf_amax.pdf
> >
> > In addition, in the same repository above, in folder https://anonymous.4open.science/r/RCGP-65CC/, you can access all the average episodic reward curves for BOHB, DEHB, and SMAC, for both the non-max-smoothed and max-smoothed experiments.
> >
> > We believe these experiments address two more concerns:
> > - “Some missing details. I couldn't find reward curves for the BOHB, SMAC, DEHB methods. Moreover, the reward curves for the proposed method but without the max smoothing were also missing. ”
> > -  “The performance for the other algorithms is not provided for the max smoothing, so it remains unclear whether the new method performs well because of the max smoothing or because of the reward modeling.”

---

> > > ### Author Response · Authors · 2022-12-01
> > > **Demanded experiments (2/2)**
> > >
> > >
> > > **On running for longer HPO budgets**:
> > >
> > > A common metric in the HPO community for evaluating the distance of the discovered configuration (a.k.a. incumbent) to the optimal configuration (a.k.a. oracle) is the regret defined as (oracle - incumbent). If we let the HPO run longer we are going to discover the oracle in the best-case scenario. In our tabular benchmark, the oracle is known (we can argmax the final returns of all possible hyperparameter configurations), therefore we can directly assess the potential gain of running longer HPO by computing the regret. Furthermore, a percentual difference variant is (oracle - incumbent)/oracle, which shows the potential gain in percentage.
> > >
> > > We present the results at https://anonymous.4open.science/r/RCGP-65CC under the folder “Normalized Regret Plots”. As can be observed the percentual regret is between 10% and 1%.
> > >
> > > Therefore, letting our HPO run till infinity will improve the results between 1% to 10% in the best-case scenario.
> > >
> > > In conclusion, we believe that this experiment addresses the concern on the gain in performance when running the HPO longer. We remain at your disposal for additional questions on this point.

---

> > > > ### Author Response · Authors · 2022-12-01
> > > > **On the experimental protocol**
> > > >
> > > > We thank the reviewer for their question to detail our evaluation protocol. We agree that a transparent and fair evaluation protocol is of the utmost importance and believe that integrating this more detailed information indeed makes the paper much clearer and stronger.
> > > >
> > > > **On the evaluation protocol**:
> > > >
> > > > We believe the comments reflect a misunderstanding of the experimental protocol, therefore, to remove any remaining doubts we detail it explicitly in the following.
> > > >
> > > >
> > > > Our evaluation protocol for each method in each environment is as follows:
> > > > ```{r, eval=FALSE}
> > > > FOR 10 seeds:
> > > >     FOR different budgets B:
> > > >         1. Search phase:
> > > >             Run HPO for a budget B and return the best hyperparameter configuration.
> > > >         2.  Evaluation phase:
> > > >             Take the best configuration returned in step 1.
> > > >             Train it for the full budget.
> > > >             Evaluate it for 10 episodes and output the mean final return of these episodes.
> > > > ```
> > > > For all Figures 1-8, we use the following procedure to generate the plots for each benchmark.
> > > >
> > > > ```{r, eval=FALSE}
> > > > FOR the same 10 seeds as above:
> > > >     FOR the same different budgets B as above in the x-axis:
> > > >         Compute the y-axis by following steps (i) and (ii):
> > > >         (i). For each environment in a benchmark:
> > > >             For each method, get the final return (step 2. above).
> > > >             Compute the rank of each method for that environment based on the returns.
> > > >         (ii). Compute the mean of the ranks across the environments in (step (i)) for each method.
> > > > ```
> > > > Ultimately, we have mean ranks across environments (step (ii)), as well as a variance due to the 10 seeds.
> > > >
> > > > **Code for postprocessing and plotting**
> > > >
> > > > To avoid any confusion, we now provide the code used to generate the plots at https://anonymous.4open.science/r/RCGP-65CC/source/plot_all_one_by_one_rcgp.py
> > > >
> > > >
> > > > **As a result, after re-summarizing the exact protocol, we would like to highlight that the following remarks are incorrect**:
> > > >
> > > > - “First, the evaluation is only based on the performance during HPO with a policy that is not fully trained.”
> > > >
> > > > We refer to step 2. of the evaluation protocol. We query from the benchmark the final return of a fully-trained policy corresponding to the best hyperparameter configuration, found by an HPO method until a budget B. The source code corresponding to this part is:
> > > > https://anonymous.4open.science/r/RCGP-65CC/source/eval_performance_from_history_gray_box.py line 43
> > > >
> > > >
> > > > - “I believe they take the max in this history, and take that as the evaluation of the algorithm.”.
> > > >
> > > > Again, we refer to the protocol above, step 2. We do not use the max in history as the evaluation of any HPO algorithm. The specific section in the source code is at:
> > > > https://anonymous.4open.science/r/RCGP-65CC/source/eval_performance_from_history_gray_box.py lines 49-51.
> > > >
> > > >
> > > > Given that we assess your comments to reflect a misunderstanding of our protocol, we would be glad to know whether we alleviated your doubts. Please feel free to ask for any clarification or a specific location at the code.
> > > >
> > > > We believe to have followed rigorously the standard and the methodologically correct protocol for comparing HPO methods in RL.
> > > >
> > > > We are willing to clarify our experimental design in even more detail if needed.

---

> > ### Comment · Reviewer_eXNC · 2022-12-03
> > **Update: increased score**
> >
> > Thank you for the additional details and experiments below. Based on these, I have increased my score to 6. I have added a short discussion of the update into my review under the "Update 2" section.
> >
> > I would suggest using the standard error instead of the standard deviation for your error bars, i.e. divide the standard deviation by sqrt(num_samples), i.e., in your case by sqrt(10) as there are 10 seeds. This way it will be easier to judge the statistical significance by just looking at the plots. (Shrinking the errorbars will also simply make the figures easier to look at.)
> >
> > I would be happy to see the work published as long as the new results and explanations are incorporated into the paper.

---

### Official Review · Reviewer_h99m · 2022-10-25

**Confidence:** 4
**Correctness:** 3
**Technical Novelty And Significance:** 3
**Empirical Novelty And Significance:** 2
**Recommendation:** 6

**Clarity, Quality, Novelty And Reproducibility:**

The paper is mostly well written. The quality of the experiments section could be improved (see comments above). A few minor issues are listed below.

* How is "rank" defined in the experimental comparisons?

* How long was full training routine for the tasks in Fig. 5 and 6? The time scales in these plots only go to $10^6$ steps, instead of $10^7$ as in the previous ones.

* Which of the three Mujoco environments is referred to as "MUJOCO" in the plots? Is it an average result over them?

* The acquisition function for the GP-BO baseline is not stated in the main text, nor the appendix. There are also no detail on how hyper-parameters are tuned for the other baselines.

Typos:
* Sec. 3: ``We define the history of $N$ evaluated configurations'', $N$ or $K$ (as in the rest of the paragraph)?
* Eq. 2: Definition of $p_H$ is missing. Also, I believe the $\log$ term is incomplete. Shouldn't it be the log of a probability (density)? If so, the $p$ is missing.


**Strength And Weaknesses:**

### Strengths
* The idea of using reward-curve models to inform multi-fidelity kernel functions for HPO is interesting.
* Experimental evaluation has a sufficient number of algorithmic baselines.
* The proposed algorithm seems relatively simple to implement compared to other baselines while achieving high performance in the low-budget regime.

### Weaknesses
Most of my concerns regard the experimental evaluation, which seems to be the main focus of the paper.

* **Number of trials** Experiments were repeated for only 3 independent trials (3 random seeds). This number seems to be a bit too small for state-of-the-art claims. DEHB, for example, had 50 trials for most experiments (Awad et al., 2021)

* **Baselines configuration** All of the HPO baselines have their own hyper-parameters and architectural choices (e.g., kernel, acquisition function, etc.). How were these configured to provide a fair comparison? Given the gap of around one std. deviation (out of only 3 random seeds), it might be possible to fine tune the baselines to obtain better results with them.

* Why use the maximum budget for the single-fidelity baselines? With 4 initial configurations provided, if I understood it correctly, it leaves them with only 6 attempts available to try to find anything. Meanwhile RCGP would have up to 10 times more iterations to run for with lower-fidelity observations.

**Summary Of The Paper:**

This paper presents a Bayesian optimisation framework for tuning hyper-parameters reinforcement learning algorithms. The framework is based on a kernel formulation which uses a model reward learning curves to interpolate between multiple budget-dependent fidelity levels. An extensive experimental evaluation is presented comparing the proposed approach against existing hyper-parameter optimisation (HPO) baselines on problems with popular RL algorithms and multiple environments.

**Summary Of The Review:**

The paper's contributions are novel, but the experiments section needs improvement and further clarifications. Otherwise, it imposes the question that the gains might be only marginal or fruit of the stochastic nature of the problems, given the low number of independent trials (3) and lack of clarity about the tuning of the baselines.


#### Post-rebuttal update
I've read the authors response and decided to raise my score (5 to 6).

---

> ### Author Response · Authors · 2022-11-19
> **Initial Response**
>
> **On the considered number of seeds**: By now we have increased the number of seeds from 3 to 10 to give more evidence on how our method performs. For the performance across 10 seeds, please see the updated Figures 1-6, and 9-11. Additionally, please note that we invested our available resources in performing experiments that take into account multiple different configuration scenarios. This allows us to consider a wider view on how our proposed sample-efficient HPO method performs in all facets of reinforcement learning. Thus we believe that our study is very valuable to the broader RL community, rather than a smaller subset.
>
> **On baselines and their configurations**: We are strong proponents for tuning algorithms. However, please note that considering tuning of HPO methods here would be unrealistic. Whenever a user employs an HPO method, they will not spend their available, limited resources on first tuning the HPO method but rather directly use the HPO method out-of-the box. Thus, in the AutoML community it has become standard practice to consider default settings of HPO methods when comparing their performances. Following this standard, we set up our experiments to take this into account. Further, please also note that we consider a broad variety of HPO methods to provide as much insight on our methods' performance as possible. This has often been neglected when novel HPO methods tailored to RL have been proposed. This left the burden of figuring out how the novel method compares to existing methods to the RL practitioners.
>
> When we consider single-fidelity baselines, we only consider the maximum budget as such methods are designed to use the highest fidelity. Such methods have no mechanism that enables exploitation of insights that could be gained on lower-fidelities. Therefore, it would be detrimental and effectively be very much like random search, rather than informed search.
>
> **How is “rank” defined?**: The rank is a metric that takes into account if a method outperformed another. Thus, a winner would have rank 1, the second best method rank 2 and so on. For our plots, we do this comparison at various stages of the optimization procedure. From our plots we can thus learn if a method is particularly well suited for different stages of the optimization procedure. For example, Figure 3 of our experiments shows that RCGP consistently outperforms the considered multi-fidelity baselines on average. Out of these, SMAC performs worst in the beginning stages but outperforming BOHB (and sometimes DEHB) in later stages.
>
> **How long was full training routine for the tasks in Fig. 5 and 6?**:
> Figure 5 and 6 compare the performance of PBT, PB2, and our method. As we describe in the Experimental Setup section, we run PBT and PB2 with four workers each, that run in parallell for the length of one full training routine, which in our setup is 10^6 steps in the training environment. Thus, for a fair comparison, we also run RCGP for a budget of four full training routines, which is the budget we show in Figures 5 and 6.
>
> **Which of the three Mujoco environments is referred to as "MUJOCO" in the plots?**:
> The performance shown in the plots with the title “MUJOCO” represents the average ranking of the respective methods across all MuJoCo environments included in AutoRL-Bench:  Ant, Hopper, and Humanoid. We include a full list of the environments and their respective type of action space in Table 2 in the Appendix.
>
> **What is the acquisition function for the GP-BO?**: We used the commonly chosen EI (expected improvement) acquisition function. We did not tune this choice as it is standard in the AutoML community to consider default configurations for HPO methods.

---

### Official Review · Reviewer_Etq6 · 2022-10-25

**Confidence:** 3
**Correctness:** 4
**Technical Novelty And Significance:** 4
**Empirical Novelty And Significance:** 4
**Recommendation:** 6

**Clarity, Quality, Novelty And Reproducibility:**

High clarity: Well written with excellent related work context. Most details are in the paper, but those that did not fit (e.g. BO, deep kernel learning, etc.) are referred to suitable related work.
High quality and seemingly high novelity.
The method is well described and source code is available (I did not test it).

**Strength And Weaknesses:**

Strength
========

* The context of the work is well described and the approach is well motivated given related works.
* The approach is clever and attack the core problem of sample-efficient HPO for RL, namely to predict the continuation of reward curves given the training epochs so far. The connection to and use of multi-fidelity GP well made and highly suitable.
* The evaluation is strong, with a variety of methods and environments.
* Clear hypotheses, and interpretation of results in light of these.
* The ablation study enriches the contributions.

Weaknesses
========
Nothing obvious

**Summary Of The Paper:**

The authors propose a new approach to HPO tailored towards RL. Specifically, a HPO approach that is sufficiently sample efficient such that the approach is effective even with limited resources (research lab as opposed to data center) is sought after. Bayesian optimization is used, with a Gaussian process (GP) surrogate. The GP has a novel deep kernel which consists of a parametrized Richard's curve, where the five coefficients of the curve are what the trained deep neural network is used to predict. The purpose of the kernel is to represent, and sample-efficiently infer, reward curves for specific hyperparameter configurations. In an extensive evaluation, on several problem domains comparing with several different kinds of state-of-the-art HBO methods, the proposed approach is demonstrated to be highly effective and clearly outperform competing methods.

**Summary Of The Review:**

The paper is well contained, well positioned and makes seemingly large and significant advances on an important problem for RL in general. It was a pleasure to read.

---

### Update
Thank you for engaging with the reviewers and clarifying many important things. However, after reading through the other reviews I reduce my score, mainly due to evaluation concerns as pointed out by h99m and eXNC. I still believe that the paper contain contributions of interest to the community, but the clarity and motivation for parts of the evaluation should probably be improved in accordance their comments. eXNC also raise important concerns in the latest comment & update, which affect my judgement. I am consequently still in favor of accept, but only marginally.

---

> ### Author Response · Authors · 2022-11-19
> **Initial Response**
>
> Thank you very much for the kind words about our work. Indeed, we set out to tackle the problem of sample-inefficiency of HPO for RL. We hope that this work will enable future breakthroughs in novel application areas as potential users of RL do not need to be experts on how each hyperparameter affects the downstream problem. Our improved experiments provide further evidence that our method is capable of doing HPO in a sample-efficient manner, on a scale at which common research labs can operate.

---

> ### Author Response · Authors · 2022-12-05
> **Thanks for the update**
>
> Thanks for the encouraging remark on our engagement with the reviewers and for investing time in reading our rebuttal.
>
> As you reduced your score due to the original criticisms of reviewer eXNC, we would like to inform you that we now address these criticisms in our comment (https://openreview.net/forum?id=rmoMvptXK7M&noteId=oCODxGq0VO) and reviewer eXNC was satisfied by this, raising their score from 3 to 6.
>
> If you have any further questions or concerns we would be pleased to discuss them.

---

### Author Response · Authors · 2022-11-19
**General Response**

Dear reviewers,

we thank you for taking the time to provide helpful feedback. In the meantime we have been working on implementing your suggestions, and, in particular, we have extended our experiments as follows:
- We increased the number of seeds for all experiments to 10 providing further evidence that our method enables sample-efficient HPO for RL;
- To show the sensibility of the discretization of the search spaces, we additionally compare to PBT on a continuous PPO configuration space for the MuJoCo environments;
- We provide additional plots that show the rewards to complement the existing ranking plots.

Besides these improvements to the experiments, we address your individual comments in individual responses to each review.

---

### Comment · Area_Chair_2hu4 · 2022-12-10
**Clarification about page 4**

This did not appear in the reviewers' comments, but I wanted to share some feedback of my own with the authors.

The discussion on page 4 that introduces the novel predictive method is unclear.

It states, "In this paper, we model the reward curve R (λ, b) of configuration λ at budget b as a generalized logistic function (Richard’s curve) with five coefficients". Then it defines a function R-hat in equation (6). Then it states "we propose a novel GP that exploits the pattern of the reward curve of the RL algorithm, by introducing the sigmoidal reward curve of (Equation 6)." It then defines a kernel, which I believe is the kernel of this GP, in equation (7). This kernel takes as input R-hat.

My understanding is that the reward curve R (λ, b) is *not* modeled as a generalized logistic function.

Instead, R (λ, b) is modeled as a GP whose kernel takes as input a feature (R-hat(λ, b, w)) that is a generalized logistic function.

If this is true, then another piece of feedback is that this model won't necessarily have a sigmoidal relationship between R (λ, b) and b.  For example, draws from a GP are often non-monotone. I understand that the 3rd feature will be sigmoidal for each lambda, but this doesn't imply that the GP draw will be sigmoidal.

---

> ### Author Response · Authors · 2022-12-12
> **Response to the request for clarification**
>
> We thank the AC for the raised question.
>
> Bayesian optimization (BO) relies on a probabilistic performance predictor (a.k.a. surrogate) that given a hyperparameter configuration, estimates the posterior distribution of the response (e.g. return in the case of RL).
>
> In our case, our BO surrogate is a gray-box Gaussian Process (GP) as was stated in the manuscript (e.g. section 3.1).
>
> However, compared to a vanilla gray-box GP, we add an auxiliary feature, the estimation of the return using a generalized logistic function. This feature is motivated by our prior knowledge of RL curves' patterns and offers a piece of auxiliary context information that conditions the GP's predictions.
>
> Therefore, we model the reward curve using a logistic function (R hat in Eq. 6). However, that model is not directly the BO surrogate, but serves as an auxiliary feature for the surrogate.
>
> The comment of the AC that the GP will not necessarily have a sigmoidal relationship is correct, however, we did not state at any point that the GP itself has an explicit logistic relationship. In contrast, we specifically clarified that it **includes/exploits** the logistic model (Eq. 6) as an auxiliary feature:
>
> Section 1: "We combine this insight in a novel gray-boy Bayesian optimization method that **includes** a parametric reward curve extrapolation layer in a neural network for computing a Gaussian process kernel."
>
> Section 3.1: "We **augment** the [GP] feature space with the estimation of the reward curve". Eq. 7 and the paragraph below clarify the mechanism.
>
> Furthermore, we provide ample empirical evidence on the benefit of adding the logistic model as a feature for a gray-box GP in the ablation of Section 5.1 (Figures 7 and 8). The gain of the green line over the blue one is the impact of the logistic model feature.
>
> A quick question a reader might raise is: Why don't you use Eq. 6 as the surrogate directly, instead of plugging the prediction into a GP? The reason is simple, we need a probabilistic surrogate model to conduct Bayesian optimization (BO) because BO needs the posterior distribution to explore configurations. The GP is required due to its posterior distribution, while Eq. 6 is a non-probabilistic point-estimate model.

---

> > ### Comment · Area_Chair_2hu4 · 2022-12-12
> > **Re: Response to the request for clarification**
> >
> > Thank you for the clarification.

---

### Author Response · Authors · 2022-12-12
**Updated manuscript**

We thank again the reviewers for their precious time.

We recently integrated all the new experiments and discussions of the rebuttal into the manuscript.

Specifically, we:

i.  added the new experimental results on the predictive accuracy of our surrogate as Appendix B.6 and referenced it from Section 3.1

ii. added the ablation experiments on comparing to the multi-fidelity HPO baselines that also use max-smoothed reward curves in Appendix B.4, referencing it from Section 5, Hypothesis 2.

iii. added the regret analysis of running the HPO for longer budgets in Appendix B.5 and referenced it from Section 5, Hypothesis 2.

iv. reworked the description of the experimental protocol in section 4.1 and pointed to the detailed descriptions of the protocol in Appendix C, including the pseudocode for plotting results in Appendix D.

The new manuscript version is accessible at the following anonymous repo:

https://anonymous.4open.science/r/RCGP-65CC/Gray_Box_Gaussian_Processes_for_AutoRL.pdf

---

### Decision · Program_Chairs · 2023-01-20

**Decision:**

Accept: poster

**Justification For Why Not Higher Score:**

- Weak methodological novelty
- Lack of clarity, particularly when describing the aspects of the methodology that is novel, but also throughout Section 3
- Some lingering questions about the experimental evaluation

**Justification For Why Not Lower Score:**

- Important problem
- Method outperforms recently published benchmark methods on a wide variety of problems

**Metareview: Summary, Strengths And Weaknesses:**

This paper proposes a new hyperparameter optimization (HPO) method for RL algorithms.

It takes advantage of the ability to run an RL algorithm using a smaller number of iterations than the algorithm requires to achieve its full performance. This produces a trace of the reward (the "learning curve") that can be extrapolated to quickly judge the quality of a set of hyperparameters. To take advantage of this, the paper proposes a novel kernel for GP regression that can extrapolate learning curves. It uses this with the expected improvement acquisition function within a Bayesian optimization algorithm. SOTA performance is demonstrated on problems from a collection of RL environments using 3 benchmark methods from the recent literature on HPO.

Specifically, the novel aspect of the kernel (equation 7) is that it uses an additional feature (equation 6), which is a Richard's curve whose parameters are given by a multi-layer perceptron taking the hyperparameters as input.


Strengths
- HPO for RL is important and is understudied
- Using learning curves is important and natural in HPO for RL
- Experimental evaluation shows a good improvement over recent methods on a wide collection of problems

Weaknesses
- Novelty --- BO with learning curves is well-studied for hyperparameter optimization in supervised learning (e.g., Falkner et al. 2018 as cited in the paper) and the approach taken here is not especially novel. This is the opinion of the AC --- some reviewers were not particularly concerned about novelty.
- Incomplete and unclear justification for the methodologically novel aspect of the approach. This is described in the review that I added.
- Lack of clarity in section 3 (beyond the specific part of section 3 discussed above). For example, the specific acquisition function used is not stated in the paper.

Minor Weaknesses
- The reviewers remain concerned about the way in which error bars were produced, even after discussion with the authors
- Use of the term "gray-box" that is overly specific and restricted, as explained by reviewer 8pAg

**Note From Pc:**

if the above contains the word "oral" or "spotlight" please see: "oral" presentation means -> notable-top-5% and "spotlight" means -> notable-top-25%. As stated in our emails, we are disassociating presentation type from AC recommendations

**Summary Of Ac-Reviewer Meeting:**

There was a great deal of discussion between the reviewing team and the authors during the reviewing process. The reviewing team felt that the authors were quite helpful during this process.

Updates to the paper and author responses to reviews significantly increased the reviewing team's opinion of this paper. These updates and discussions focused on the empirical evaluation --- it included, e.g., discussion of the ranking-based approach to evaluation used in the paper, the number of seeds used, the use of single-fidelity baselines, and the way in which the maximum was taken in evaluation.

By the time the AC-reviewer meeting occurred, the reviewers were largely aligned on a rating of 6.

In the AC-reviewer meeting, we summarized strengths and weaknesses, which are articulated above. There was some difference of opinion about whether lack of novelty was a weakness of the paper, with some reviewers feeling that it is fine to use largely-standard methods (HPO for problems with learning curves) on a new problem (HPO for RL) and others (including the AC) feeling that more methodological novelty would have improved the paper.  The AC and reviewers were aligned on the other points described in the metareview. Everyone agreed that the authors did a good job during the rebuttal process, that their responses and updates to the paper were helpful, and that the main technical issues raised in the reviews had been addressed.